# Machine learning identifies T cell receptor repertoire signatures associated with COVID-19 severity

Jonathan J. Park [1,2,3,4,5,11], Kyoung A V. Lee[1,2,3,6,11], Stanley Z. Lam[1,2,3], Katherine S. Moon[1,2,3], Zhenhao Fang[1,2,3] & Sidi Chen [1,2,3,4,5,7,8,9,10 ✉]

T cell receptor (TCR) repertoires are critical for antiviral immunity. Determining the TCR repertoire composition, diversity, and dynamics and how they change during viral infection can inform the molecular specificity of host responses to viruses such as SARS-CoV-2. To determine signatures associated with COVID-19 disease severity, here we perform a large-scale analysis of over 4.7 billion sequences across 2130 TCR repertoires from COVID-19 patients and healthy donors. TCR repertoire analyses from these data identify and characterize convergent COVID-19-associated CDR3 gene usages, specificity groups, and sequence patterns. Here we show that T cell clonal expansion is associated with the upregulation of T cell effector function, TCR signaling, NF-kB signaling, and interferon-gamma signaling pathways. We also demonstrate that machine learning approaches accurately predict COVID-19 infection based on TCR sequence features, with certain high-power models reaching near-perfect AUROC scores. These analyses provide a systems immunology view of T cell adaptive immune responses to COVID-19.

[1] Department of Genetics, Yale School of Medicine, New Haven, CT, USA. [2] Systems Biology Institute, Yale University, West Haven, CT, USA. [3] Center for Cancer Systems Biology, Yale University, West Haven, CT, USA. [4] MD-PhD Program, Yale University, New Haven, CT, USA. [5] Molecular Cell Biology, Genetics, and Development Program, Yale University, New Haven, CT, USA. [6] Department of Biostatistics, Yale School of Public Health, New Haven, CT, USA. [7] Immunobiology Program, Yale University, New Haven, CT, USA. [8] Yale Comprehensive Cancer Center, Yale School of Medicine, New Haven, CT, USA. [9] Yale Stem Cell Center, Yale School of Medicine, New Haven, CT, USA. [10] Yale Center for Biomedical Data Science, Yale School of Medicine, New Haven, CT, USA. [11] These authors contributed equally: Jonathan J. Park, Kyoung A V. Lee. ✉email: sidi.chen@yale.edu

Much of the ongoing COVID-19 vaccination strategies have focused on targeting B cells for eliciting neutralizing antibodies (nAbs) against SARS-CoV-2[1,2]. However, SARS-CoV-2 nAb levels after infection or vaccination have been found to decrease over time[3], and recently emerging variants of concern (VOC) have been associated with antibody escape[4]. Strategies that solely focus on nAbs may not be sufficient for managing the pandemic in the long term. Thus, there has been increasing interest in studying the role of T cell immunity in response to COVID-19 infection[5,6].

Functional T cell responses are crucial for the control and clearance of many respiratory viral infections[7], including for SARS-CoV and MERS-CoV[8,9]. Studies from transgenic mouse models suggest that T cells are also crucial for disease resolution after infection with SARS-CoV-2[10], and SARS-CoV-2-specific CD4+ and CD8+ T cells have been associated with milder disease in human patients[11], suggesting roles for coordinated adaptive immune responses in protective immunity against COVID-19. T cells contribute to viral control through numerous mechanisms, including supporting the generation of antibody-producing plasma cells (T follicular helper cells), production of effector cytokines such as IFN-gamma and TNF, and cytotoxicity against infected cells. Generation of memory T cells can provide life-long protection against pathogens[12], and a recent study showed that SARS-CoV-2-specific memory T cell responses were sustained for 10 months in COVID-19 convalescent patients[13]. Moreover, there is mounting evidence that SARS-CoV-2 VOCs rarely escape T cell reactivity[14], perhaps partly due to a broader distribution of T cell epitopes across the entire viral proteome, unlike nAb target limitation to the viral surface. Due to the importance of T cells in long-term and broad immune reactivity, there has been an increase in diverse vaccine strategies to expand targets beyond the spike protein and induce T cell responses[5].

T cells recognize viral antigens presented on major histocompatibility complex (MHC) molecules through an enormously diverse assembly of T cell receptors (TCRs)[15]. Ligation of the TCR by peptide-loaded MHC molecules leads to T cell activation and clonal expansion, causing a shift in repertoire specificity towards the antigen. Therefore, TCR repertoires represent a functional signature of the adaptive immune response. The development of high-throughput DNA sequencing methods has enabled highly quantitative investigation into the diversity and composition of immune repertoires[16]. As TCRs are cell-specific and represent a type of molecular tag of T cells, TCR sequencing is increasingly becoming an essential tool in informing clinical understanding of disease via monitoring the dynamics of T cells in diseases. Other studies have used tracking of TCR repertoires in cancer patients over time to identify correlations between clonal dynamics and clinical features such as immunotherapy treatment response[17,18]. TCR-seq data thus has great potential for gaining quantitative insight into the patterns of adaptive immune responses, which has been particularly well demonstrated in studies for cancer immunology.

In the context of COVID-19, T cell studies have revealed preliminary insights into the adaptive immune response to COVID infection. One study reported a decrease in CD4+ and CD8+ T-cell counts and decreased T-cell clonal expansion in early recovery stage patients compared to healthy controls[19], while another study reported that there were increased proportions of active state T cell subsets in COVID-19 patients[20]. The disease-severity-dependent clonal expansion was proposed in another study, which showed increased CD8+ T cell clonal expansion in moderate COVID-19 patients compared to severe COVID-19 patients[21]. Clonality and skewing of TCR repertoires were linked to Type I and Type III interferon responses, early CD4+ and CD8+ T cell activation, and nonconventional Th1 cell polarizations[22]. In severe COVID-19 patients, profound immune exhaustion with skewed T cell receptor repertoire and broad T cell expansion was observed, while in moderate patients, an intensive expansion of highly cytotoxic T cell subsets was observed, suggesting that the expansion of appropriate T cell subsets is impaired in severe COVID-19 patients[20].

We sought to develop a systems immunology approach for investigating TCR repertoires from COVID-19 patients to help decode patterns of the adaptive immune response during SARS-CoV-2 infection. While there have been several studies on different aspects of TCR-seq analysis for COVID-19[19,20,22,23], there have been limited studies that incorporate motif-based analysis, transcriptomics, and machine learning in a large-scale, comprehensive investigation into the immune responses during disease course of varying severity. We anticipate that our approach here can provide sets of COVID-19-associated sequences and motifs that may help guide the development of prognostic and diagnostic markers and potentially help design therapeutic interventions that better harness the power of T cell immunity.

## Results
**TCR repertoires from COVID-19 patients and healthy donors reveal trends in CDR3 gene usage and diversity.** To determine if there were any global patterns that distinguish the immune repertoires of COVID-19 patients, we systematically compiled and analyzed TCR-seq samples (total $n = 2130$) from COVID-19 patients and healthy donors (Fig. 1A, Supplementary Dataset S1). TCR repertoire datasets were obtained from studies by Adaptive Biotechnologies (AB, $n = 1574$), ISB-Swedish COVID-19 Biobanking Unit (ISB-S, $n = 266$), PLA General Hospital (PLAGH, $n = 20$), and Wuhan Hankou Hospital (WHH, $n = 15$), and then uniformly processed for downstream analysis (see Methods). The individual datasets underwent separate but standardized processing using identical bioinformatic pipelines, as opposed to combining the data into one pooled set prior to analysis. All comparisons between COVID-19 and healthy donor controls were made within the individual dataset, and any reference to the findings in multiple datasets in the manuscript refers to the consensus findings of within-dataset comparisons. The analyses for the ISB-S dataset were further stratified by cell type (CD4 vs CD8), separated into ISB-S CD4 and ISB-S CD8 datasets, both of which contain TCR sequences of healthy donors and COVID-19 patients. Clonality analyses revealed that COVID-19 patient samples from the ISB-S CD4, ISB-S CD8, and WHH datasets had fewer total unique clonotypes compared to healthy donor controls in these respective datasets, demonstrating the consistency of this observation in multiple sources of TCR data (Fig. S1A). Moreover, repertoire diversity metrics, including Chao1 estimators (a measure of species richness), Gini-Simpson indices (probability of interspecific encounter), and inverse Simpson indices, were decreased for COVID-19 samples compared to healthy donor samples, notably for the AB, ISB-S CD4, and ISB-S CD8 datasets (Fig. 1B, C, S1B). The decrease in clonal diversity measures is consistent with the increase in the relative abundance of the top clonotypes in the repertoire space for COVID-19 samples (Fig. 1F, S3D, Supplementary Dataset S2), which suggests the expansion of a small number of functional clones after antigen exposure. These results together reveal global shifts in immune repertoire clonality and diversity in patients with COVID-19 compared to healthy donors. However, one intrinsic limitation to assessing TCR diversity is the fact that the dataset's size influences the diversity measurements, as the distribution of TCR frequencies is not linearly scalable. The higher variance in clonal diversity of COVID patients observed in the AB dataset (Fig. 1B, C, Supplementary Dataset S4) illustrates this limitation.

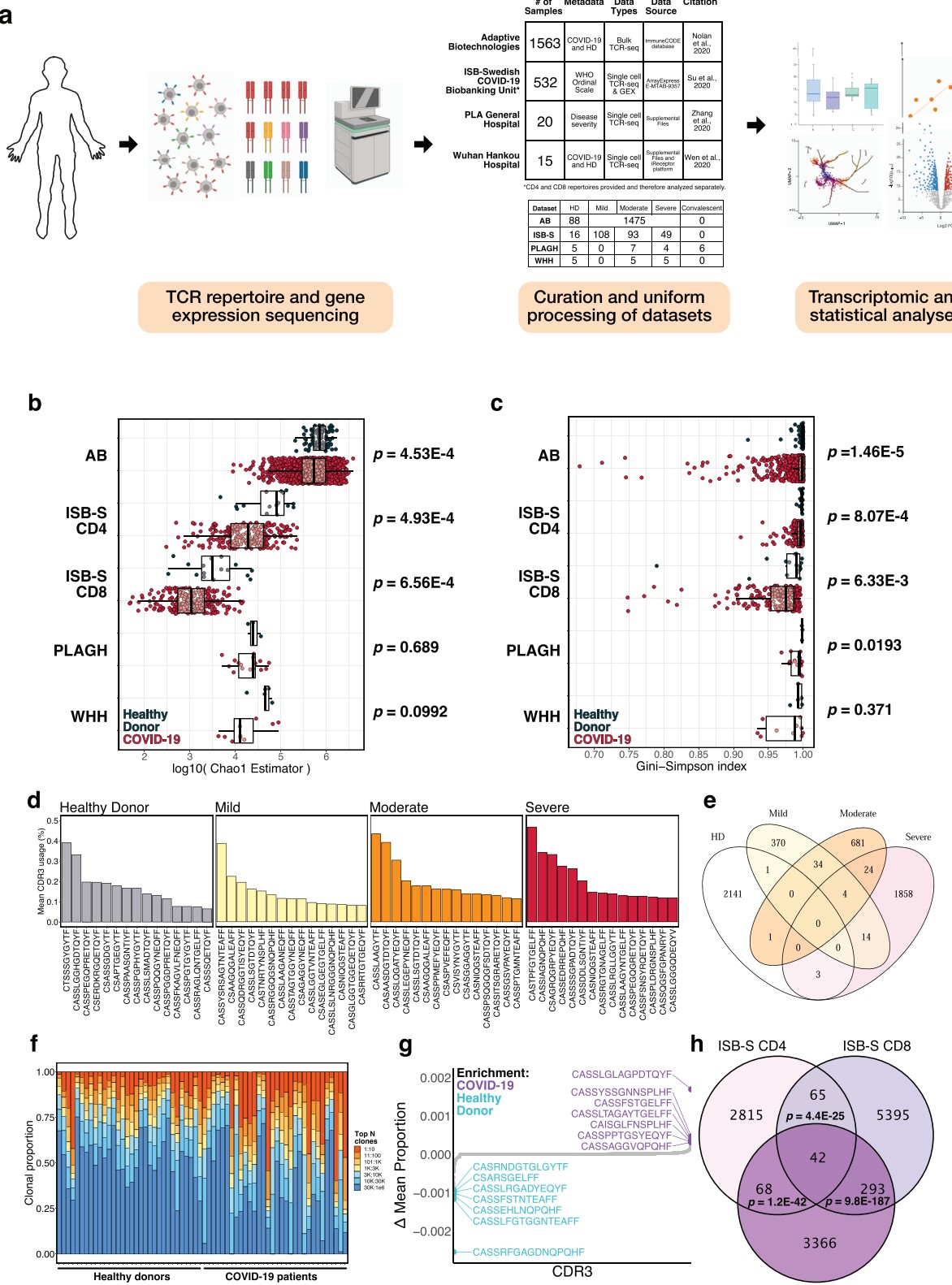

The larger spread in clonal diversity in the COVID group in the AB dataset reflects a greater heterogeneity in T cell repertoire changes upon COVID infection compared to other datasets, but the general directionality of this change is a decrease in diversity.

To determine the specific gene usage preferences and dynamics in COVID-19 patients, we performed comparative analyses of the V(D)J gene and complementarity-determining region 3 (CDR3)

gene usage for the AB and ISB-S datasets. While we observed selective V and J gene usage differences between the TCR samples of COVID-19 patients and healthy donors from the AB dataset (Fig. S1D, E, Supplementary Dataset S5, S6), fewer differences in these gene usages were observed from the ISB-S datasets (Fig. S2A–D). Moreover, there were no differences in clonotype frequencies by CDR3 length across the datasets (Fig. S1C,

**Fig. 1 Analysis of TCR repertoires from COVID-19 patients and healthy donors reveal trends in CDR3 gene usage and diversity. a** Schematic detailing curation and analysis of TCR repertoire datasets from healthy donors and COVID-19 patients. Sequencing data was obtained from Adaptive Biotechnologies (AB, $n = 1574$), ISB-Swedish COVID-19 Biobanking Unit (ISB-S, $n = 266$, CD4 and CD8 repertoires), PLA General Hospital (PLAGH, $n = 20$), and Wuhan Hankou Hospital (WHH, $n = 15$). Created with BioRender.com. **b** Boxplot of Chao1 indices for COVID-19 patients and healthy donors for each repertoire dataset. *P*-values were obtained using the two-sided Wilcoxon rank sum test. **c** Boxplot of Gini-Simpson indices for COVID-19 patients and healthy donors for each repertoire dataset. *P*-values were obtained using the two-sided Wilcoxon rank sum test. **d** Bar plots showing the top 15 mean CDR3 usages for patients in the ISB-S CD4 dataset grouped by disease severity (healthy donor = 16, mild = 108, moderate = 93, severe = 49). **e** Venn diagram showing overlap of top mean CDR3 usages (proportion threshold = 0.0001) for patients in the ISB-S CD4 dataset grouped by disease severity. The CDR3 sequences enriched in the COVID patients have overlap among mild, moderate, and severe patients, while minimal overlap is observed between healthy donors and COVID-19 patients. **f** Bar plot depicting relative abundance for groups of top clonotypes for a random sample of repertoires (healthy donors = 32, COVID-19 = 32) from AB dataset, with relative overrepresentation of specific clonotypes in COVID-19 patients. **g** Dotted waterfall plot of CDR3 gene usage differentials between COVID-19 patients and healthy donors (delta mean proportion) in AB dataset. Purple dots are CDR3 sequences enriched in COVID-19; light blue dots are CDR3 sequences enriched in healthy donors; gray dots are all other CDR3 sequences. **h** Venn diagram showing overlap of COVID-19 enriched CDR3 sequences for patients in the ISB-S CD4, ISB-S CD8, and AB datasets (thresholds 0.0001 for ISB-S samples, 0.00001 for AB samples). *P*-values for overlap significance calculated using hypergeometric test.

Supplementary Dataset S3). By comparison, the top CDR3 sequences of healthy donors, mild COVID patients, moderate COVID patients, and severe COVID TCR patients' repertoires were different within the AB dataset, as well as within the ISB-S datasets (Fig. 1D, S3A, B). To identify COVID-19-associated CDR3 sequences that are conserved across disease conditions and datasets, we performed a series of set analyses using sequences above a proportion threshold (0.0001 for ISB-S samples, 0.00001 for AB samples, where the different proportion thresholds were applied to account for the AB dataset's relative size compared to the ISB-S dataset) for each condition. We found that the CDR3 sequences enriched in COVID patients had overlap among mild, moderate, and severe patients, while there was almost no overlap in CDR3 enrichment between healthy donors and COVID-19 patients in both CD4 T cell (Fig. 1E) and CD8 T cell datasets (Fig. S3B). Moreover, we observe 42 conserved CDR3 sequences when comparing the union set of disease-associated CDR3 sequences for ISB-S CD4 samples, the union set of disease-associated CDR3 sequences for ISB-S CD8 samples, and COVID-19 CDR3 sequences for the AB samples (Fig. 1H). In order to determine enriched CDR3 sequences for each dataset and disease conditions, we plotted the difference in mean CDR3 proportions between samples of interest and healthy donors (Fig. 1G, S3E–J). Although the identified sequences may not be definitively specific, we provide here a set of systematically processed COVID-19-associated convergent and enriched CDR3 gene usages.

**K-mer and motif analyses reveal patterns associated with disease conditions.** Sequence convergence of immune repertoires can also occur at the level of motifs, or sequence substrings, in addition to that of clones. One approach to decomposing CDR3 sequences into motifs is by using overlapping k-mers, or amino acid sequences of length k, which provide a functional representation of the repertoires with increased compatibility for statistical analyses and machine learning methods[24]. We created 3-mer, 4-mer, 5-mer, and 6-mer frequency matrix representations of ISB-S CD4 and ISB-S CD8 datasets and performed principal components analysis (PCA) to see whether samples cluster by disease severity (Fig. 2A, C, S4A–F, Supplementary Dataset S7). We found that while the majority of samples clustered together, a number of mild and moderate samples were separated from the central cluster across all analysis permutations, while severe samples are generally associated with the central cluster of samples including healthy donors. Because there are general rules that define many CDR3 amino acid sequences, such as the CASS motif commonly found at the beginning of many CDR3 sequences, it was expected that most of the data would cluster together

regardless of COVID infection status. The homogenous signal of shared CDR3 characteristics was likely to dominate the heterogenous CDR3 k-mers that differentiate individuals' repertoires. However, the outliers in the PCA that failed to fall into the large central cluster, which came from mild and moderate COVID samples, possibly indicated that the PCA is detecting high-variance data features that differentiate them from other CDR3 k-mers, while all severe COVID samples were found within the homogenous central cluster. Machine learning is a broadly useful tool to detect and identify these fine differences in biological signal. The outliers from mild and moderate COVID patient samples may suggest that T cell repertoires may be undergoing changes that selectively enrich certain clones that harbor specific TCR motifs, in response to COVID infection, which is being captured in the PCA plot. No such changes were detected in severe COVID patients' TCR motifs in the PCA. These results are consistent with emerging data that patients with severe COVID-19 have substantial immune dysregulation in comparison to those with less severe disease. Studies have shown that T cell polyfunctionality is increased in patients with moderate disease but reduced in those with severe disease[25], and there have been proposed models of TCR clonality whereby the response in mild disease includes detection of dominant clones while the response in severe disease do not[26]. Moreover, heatmaps of 3-mer abundances reveal some shared motifs between mild and moderate samples such as YNE, NEQ, EQF, and QFF for repertoires randomly sampled from the ISB-S CD4 dataset and TEA, EAF, and AFF for repertoires randomly sampled from the ISB-S CD8 dataset (Fig. 2B, D). In aggregate, these results suggest that there are sequence features that distinguish COVID-19 TCR repertoires from healthy donors to various degrees based on disease condition.

Recent sequence similarity approaches have been developed to determine TCR specificity clusters for motif-based prediction of antigen specificity and identification of key conserved residues that drive TCR recognition[27–29]. We used the Grouping of Lymphocyte Interactions by Paratope Hotspots version 2 (GLIPH2) algorithm[29] to cluster the TCR sequences based on predicted antigen specificity for motifs associated with different disease conditions in the ISB-S datasets. We also used the Optimized Likelihood estimate of immunoGlobulin Amino-acid sequences (OLGA) algorithm[30] to calculate the generation probability (pGen) of the clonotypes contained in the clusters identified from the GLIPH2 analysis. Low pGen clonotypes are considered private and not shared widely in the population, while high pGen clonotypes are considered public and shared in a large proportion of the population due to convergent recombination[22,31]. We found that the mild and moderate disease conditions had both relatively lower pGen scores and higher median frequency clusters compared to the severe disease and healthy donor

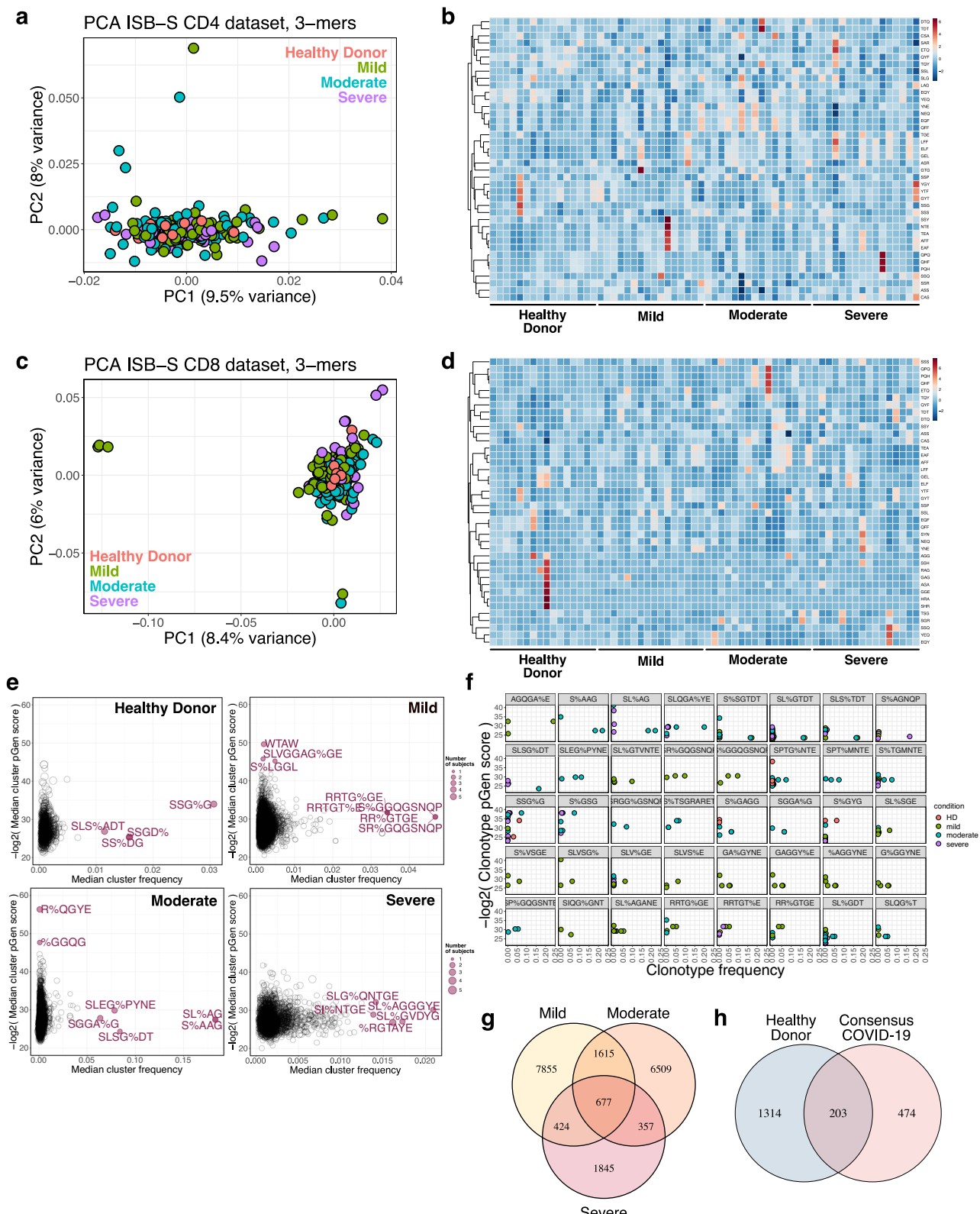

conditions for both the ISB-S CD4 and CD8 datasets (Fig. 2E, S4G, Supplementary Dataset S8–S10). Visualization of individual clusters revealed that the mild and moderate disease conditions had clonotypes with the highest proportional representation, including motifs AGQGA%E, S%AAG, SL%AG, and SLQGA%YE (the % character corresponds to a wildcard amino acid) for the ISB-S CD4 dataset (Fig. 2F) and motifs SEG%NTDT, SLDSGGA%E, SL%

SGGANE, SLAA% for the ISB-S CD8 dataset (Fig. S4H). Using set analysis within the ISB-S CD4 dataset, we found that the motif-based clustering of T cells via GLIPH2 algorithm successfully identified 677 CD4 T cell clusters that are common across all levels of severity of COVID-19 (Fig. 2G). Among these 677 CD4 T cell clusters, 474 were exclusive to COVID-19 and not found in healthy donors (Fig. 2H). Similarly, for the ISB-S CD8 dataset, we found 51 consensus clusters,

**Fig. 2 K-mer and motif analyses reveal patterns associated with disease condition. a** Principal components analysis of 3-mer representations of TCR repertoires from the ISB-S CD4 dataset ($n = 266$). **b** Heatmaps of 3-mer abundances of a random sample of repertoires from the ISB-S CD4 dataset by disease condition (healthy donor = 16, mild = 16, moderate = 16, severe = 16). **c** Principal components analysis of 3-mer representations of TCR repertoires from the ISB-S CD8 dataset (healthy donor = 16, mild = 108, moderate = 93, severe = 49). **d** Heatmaps of 3-mer abundances of a random sample of repertoires from the ISB-S CD8 dataset by disease condition (healthy donor = 16, mild = 16, moderate = 16, severe = 16). **e** Median frequency and pGen scores of COVID-19 and healthy donor associated T cell clusters from GLIPH2 analysis of the ISB-S CD4 dataset, grouped by disease condition. **f** Detailed view of frequencies and pGen scores of specific clonotypes associated with high frequency T cell clusters from CD4 dataset. Clonotypes are colored by patient disease condition. **g** Venn diagram showing COVID-19-associated T cell clusters for patients in the ISB-S CD4 dataset grouped by disease condition. 677 TCR specificity clusters were found in common across different severities of COVID-19. **h** Venn diagram showing overlap between consensus COVID-19-associated T cell clusters (taken from intersection of disease conditions in Fig. 1G) and healthy donors for repertoires in the ISB-S CD4 dataset. Among the 677 T cell clusters commonly found across all levels of COVID-19 severity, 474 clusters were exclusive to COVID-19 patients and not found within healthy donors.

35 of which were exclusively found in COVID-19 (Fig. S4I, J). We provide here all identified clusters and motifs with associated CDR3 sequences, V gene usage, and J gene usages, along with clonotype pGen scores and the identified COVID-19-associated clusters.

**Transcriptional signatures of clonal expansion and associations with disease severity.** In order to investigate the relationship between the enriched clonotypes and their transcriptomes, we performed dimensionality reduction on 137,075 CD4 T cell single-cell RNA sequencing samples that had CDR3 sequences associated with identified GLIPH2 clusters. The transcriptomes were projected to a two-dimensional space by uniform manifold approximation and projection (UMAP) (Fig. S5A). Clustering was performed using the Louvain algorithm, revealing 12 clusters with differentially expressed gene signatures (Fig. 3A, S5C). Overall, a stark contrast was observed in the clustering patterns of cells from healthy donors and COVID-19 patients in the UMAP, where cells from healthy donors were concentrated in clusters 3 and 4, while the cells from COVID patients were mainly found in cluster 6. Cluster 6 contained the proliferating subset of T cells with high degrees of clonality (Fig. 3A, S5B), suggesting phenotypic correlates of clonal expansion. Moreover, a high density of cells in cluster 6 contained the top COVID-enriched TCR sequence motifs identified from GLIPH2 motif analysis, such as AGQGA%E, S%AAG, SL%AG, SLQGA%YE, S%SGTDT, SL% GTDT, SLS%TDT, and S%AGNQP (Figs. 2F, 3C). These clonally expanded cells containing COVID-enriched TCR sequence motifs highly expressed the gene *GNLY*, which encodes the cytotoxic granules of T cells, indicating that the cells in cluster 6 are primarily activated, proliferating cytotoxic T cells. We also found a correlation between clonotype expansion and COVID infection, with cells from COVID-19 patients exhibiting the highest density in effector phenotype-associated cluster 6, while healthy donor cells exhibited density in the naïve phenotype-associated clusters (Fig. 3B). We also found a higher association of lower pGen score, or private, clonotypes with cluster 6 compared to the high pGen score clonotypes (Fig. 3D, Supplementary Dataset S10), suggesting that these clones may be specific. However, a comparison of the proportion of cells for each disease condition in cluster 6 with healthy donors revealed cell proportion increases only for the moderate condition (Fig. 3E), despite increasing trends for all conditions. In contrast, the naïve cell subset in the UMAP plots indicated by the gene markers *TCF7* and *LEF1*, were most abundant among healthy donors' T cells in clusters 3 and 4, whereas few naïve T cells were observed in cluster 6 (Fig. 3B, S5D). Altogether, these results demonstrate relationships between clonal expansion, disease status, and cell phenotype, which can be extended to subsequence motifs.

We extended this analysis to the CD8 dataset to see if the associations between clonal expansion and disease severity are

maintained. UMAP projection of 70,237 CD8 T cell single-cell transcriptomes and clustering revealed 15 clusters with differentially expressed gene signatures (Fig. 4A, S6A, S6C, Supplementary Dataset S11–S13). As with the CD4 dataset, we found clustering of cells with high degrees of clonality, distributed here across the clusters 0, 2, 3, 5, 7, 9, 10, 13, and 14 (grouped together as Expanded for further analysis) (Fig. 4A, S6B). We also found high density of top enriched GLIPH2 motifs in the Expanded group, including SEG%NTDT, SLDSGGA%E, SL%SGGANE, SLAA%, SQT%STDT, SP%SGSYE, SPGT%GYNE, and S% RQGAGGE (Fig. 4C, S4H). We observe a relatively higher density of cells from COVID-19 disease conditions in the Expanded group as compared to the healthy donors (Fig. 4B), with a low density of disease-associated cells in the non-Expanded clusters. Likewise, we found a more exclusive association between lower pGen score clonotypes and the Expanded group, particularly cluster 9 (Fig. 4D). A comparison of the proportion of cells for each disease condition in the Expanded group with healthy donors reveals cell proportion increases for all conditions (Fig. 4E). These results highlight the relationship between clonal expansion and disease severity, which is comparable to the results from the CD4 dataset.

To investigate the gene expression changes that occur with clonal expansion, we performed differential expression (DEX) analysis between cluster 6 cells versus all other cells for the CD4 dataset (Fig. 3F) and the Expanded group cells versus all other cells for the CD8 dataset (Fig. 4F, Supplementary Dataset S14, S15). Using a threshold of $q$-value < 1e-4, we found 512 downregulated genes and 959 upregulated genes for the CD4 T cell DEX, as well as 600 downregulated genes and 859 upregulated genes for the CD8 T cell DEX. Volcano plots for both T cell types revealed upregulation of cytotoxicity-associated transcripts such as granzymes and granulysin and downregulation of naïve phenotype associated markers such as *TCF7* and *LEF1*. Comparison of UMAPs of the individual subpopulation phenotype markers also showed a correlation between cluster 6 or the Expanded group clusters and effector-related markers such as *GZMA, PRF1, NKG7*, and *GNLY*, with downregulation of naïve-related markers such as *TCF7* and *LEF1* (Figs. S5D, S6D). Functional gene annotation analysis with DAVID[32] revealed enriched pathways terms such as TCR signaling pathway, regulation of immune response, NF-kB signaling, IFN-gamma mediated signaling, and TNF-mediated signaling pathways were upregulated in clonally expanded clusters (Figs. 3H, 4H, Supplementary Dataset S16) while terms such as translational initiation, viral transcription, translation, and ribosomal subunit assembly were downregulated (Figs. 3G, 4G) for both CD4 and CD8 differential expression analyses. Therefore, we find that clonally expanded CDR3 sequences and motifs are highly associated with effector T cell phenotypes at both the individual gene and functional pathway levels while downregulating a number of mRNA processing-related programs.

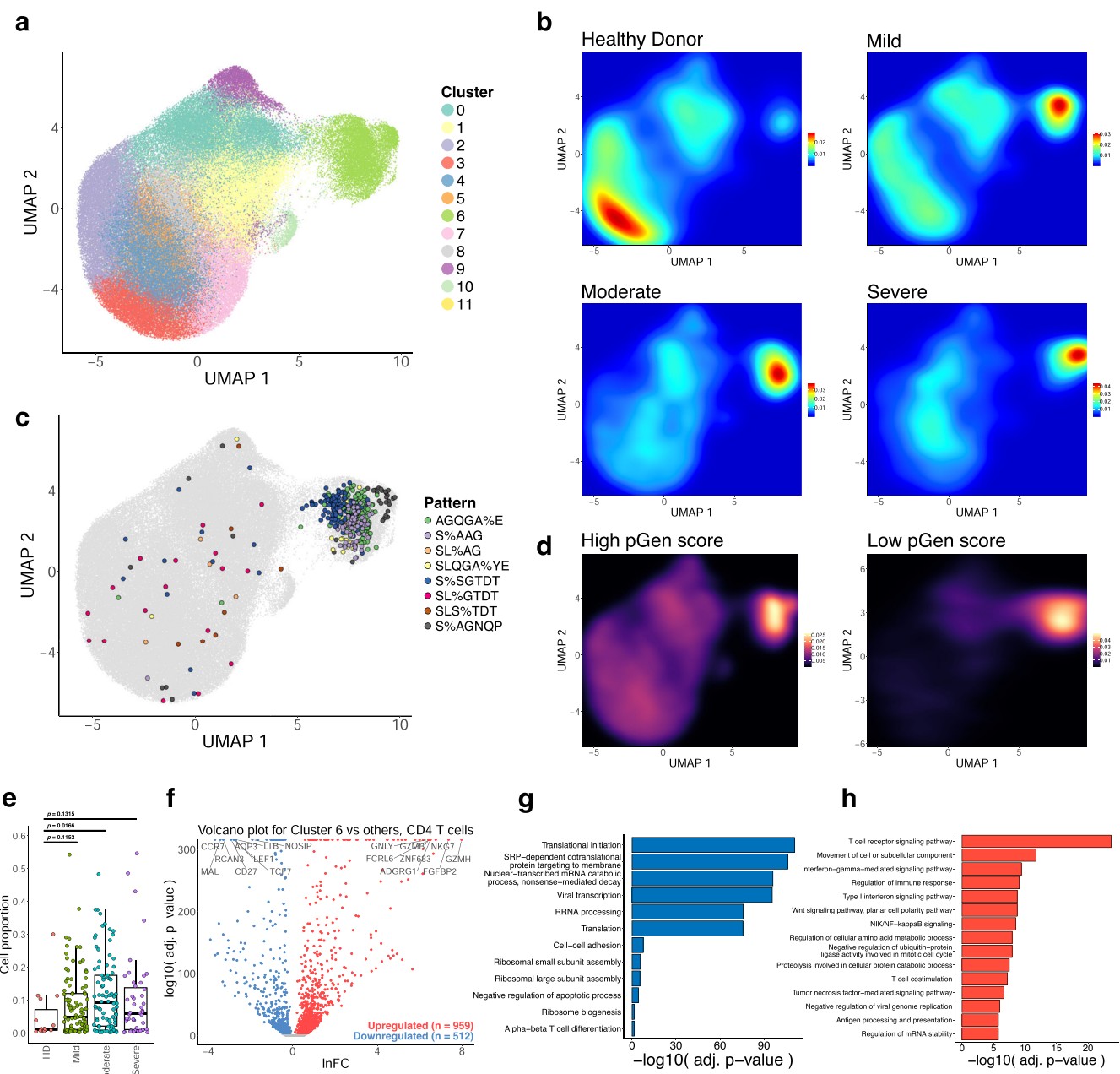

**Fig. 3 Single-cell transcriptional signatures of clonal expansion of CD4 T cells. a** UMAP visualization of 137,075 CD4 T cell single-cell transcriptomes from the ISB-S CD4 dataset pooled across samples and conditions. 12 clusters identified using the Louvain algorithm. **b** Two-dimensional density plot of cells from each disease condition (healthy donor, mild, moderate, severe) by UMAP coordinates. Red represents areas of high density of cells of a given condition; blue represents areas of low density. **c** UMAP visualization with cells labeled by top eight most frequent CD4 TCR clusters identified by the GLIPH2 analysis. **d** Two-dimensional density plot of cells with high or low pGen score clonotypes by UMAP coordinates. Yellow represents areas of high density of cells; black represents areas of low density. **e** Boxplots of clonally expanded cell proportions (in cluster 6) for each disease condition (cell count healthy donor = 544, mild = 3568, moderate = 5012, severe = 2336). Comparison between groups performed with two-sided Wilcoxon rank-sum test. **f** Volcano plot of differentially expressed genes between clonally expanded cells and all other cells in the ISB-S CD4 dataset (Cluster 6 cells = 5000, all other cells = 5000). Differential gene expression was performed with Seurat using the two-sided Wilcoxon rank-sum test; the Bonferroni corrected adjusted *p*-values and log fold-change of the average expression were used for visualization. **g** Bar plot of biological processes (BP) pathway terms associated with downregulated genes (clonally expanded cells vs all other cells, *q*-value < 1e-4) by DAVID analysis. **h** Bar plot of biological processes (BP) pathway terms associated with upregulated genes (clonally expanded cells vs all other cells, *q*-value < 1e-4) by DAVID analysis.

**Machine learning models for disease severity.** To determine whether the constitutive sequence motifs in the CDR3 sequence of the TCR contain sufficient information to be predictive of COVID-19 infection, we trained several classical supervised machine learning (ML) algorithms on the repertoires from the ISB-S CD4 and ISB-S CD8 datasets. We implemented Random Forests (RF), Support Vector Machines (SVM), Bernoulli Naïve Bayes (BNB), Gradient Boosting Classifiers (GBC), and K-Nearest Neighbors (KNN) on frequency matrices of overlapping 3-mer or 6-mer amino acids adapted from the TCR repertoires. ML models were trained as binary classification tasks to predict mild, moderate, or severe COVID-19 TCR repertoires from healthy donor

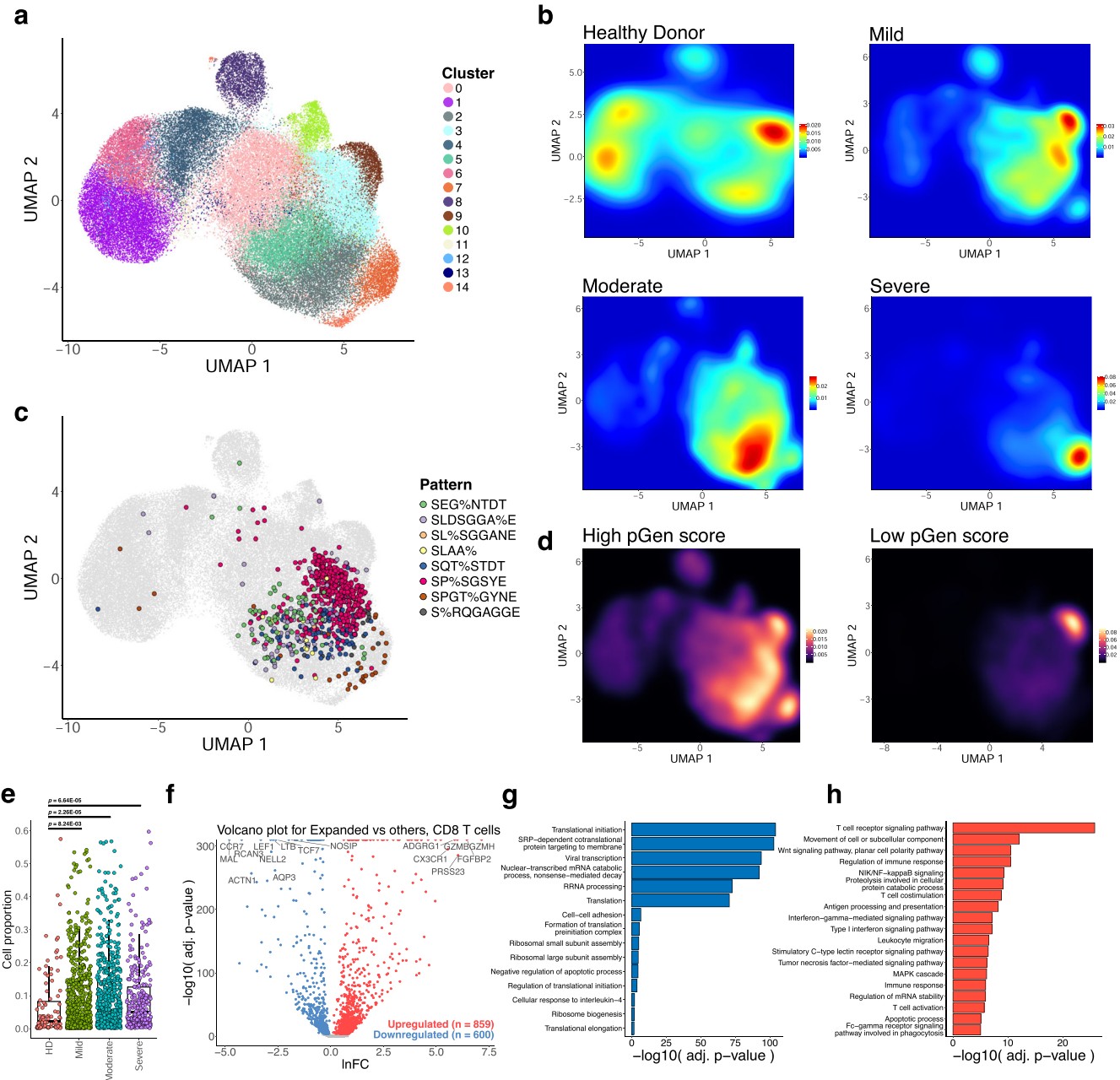

**Fig. 4 Single-cell transcriptional signatures of clonal expansion of CD8 T cells. a** UMAP visualization of 70,237 CD8 T cell single-cell transcriptomes from the ISB-S CD8 dataset pooled across samples and conditions. 15 clusters identified using the Louvain algorithm. **b** Two-dimensional density plot of cells from each disease condition (healthy donor, mild, moderate, severe) by UMAP coordinates. Red represents areas of high density of cells of a given condition; blue represents areas of low density. **c** UMAP visualization with cells labeled by top eight most frequent CD8 TCR clusters identified by the GLIPH2 analysis. **d** Two-dimensional density plot of cells with high or low pGen score clonotypes by UMAP coordinates. Yellow represents areas of high density of cells; black represents areas of low density. **e** Boxplots of clonally expanded cell proportions (in Expanded group) for each disease condition (cell count healthy donor = 2579, mild = 18,622, moderate = 15,743, severe = 7159). Comparison between groups performed with two-sided Wilcoxon rank-sum test. **f** Volcano plot of differentially expressed genes between clonally expanded cells and all other cells in the ISB-S CD8 dataset (Expanded group cells = 5000, all other cells = 5000). Differential gene expression was performed with Seurat using the two-sided Wilcoxon rank-sum test; the Bonferroni corrected adjusted p-values and log fold-change of the average expression were used for visualization. **g** Bar plot of biological processes (BP) pathway terms associated with downregulated genes (clonally expanded cells vs all other cells, q-value < 1e-4) by DAVID analysis. **h** Bar plot of biological processes (BP) pathway terms associated with upregulated genes (clonally expanded cells vs all other cells, q-value < 1e-4) by DAVID analysis.

repertoires for either CD4 or CD8 ISB-S datasets. A total of 12 models were trained, with the permutations varying in (1) classifying different levels of COVID severity (HD vs Mild, HD vs Moderate, HD vs Severe), (2) CD4 vs CD8 T cell receptors, and (3) 3mer vs 6mer representation of the TCR data. Training and testing sets were partitioned with an 80:20 ratio, then for 500 iterations, the algorithms were trained on a random 80% of the training set and evaluated for performance on the test set. We found that RFs, GBCs, and SVMs generally had strong classification performance across the board, compared to KNNs or BNB.

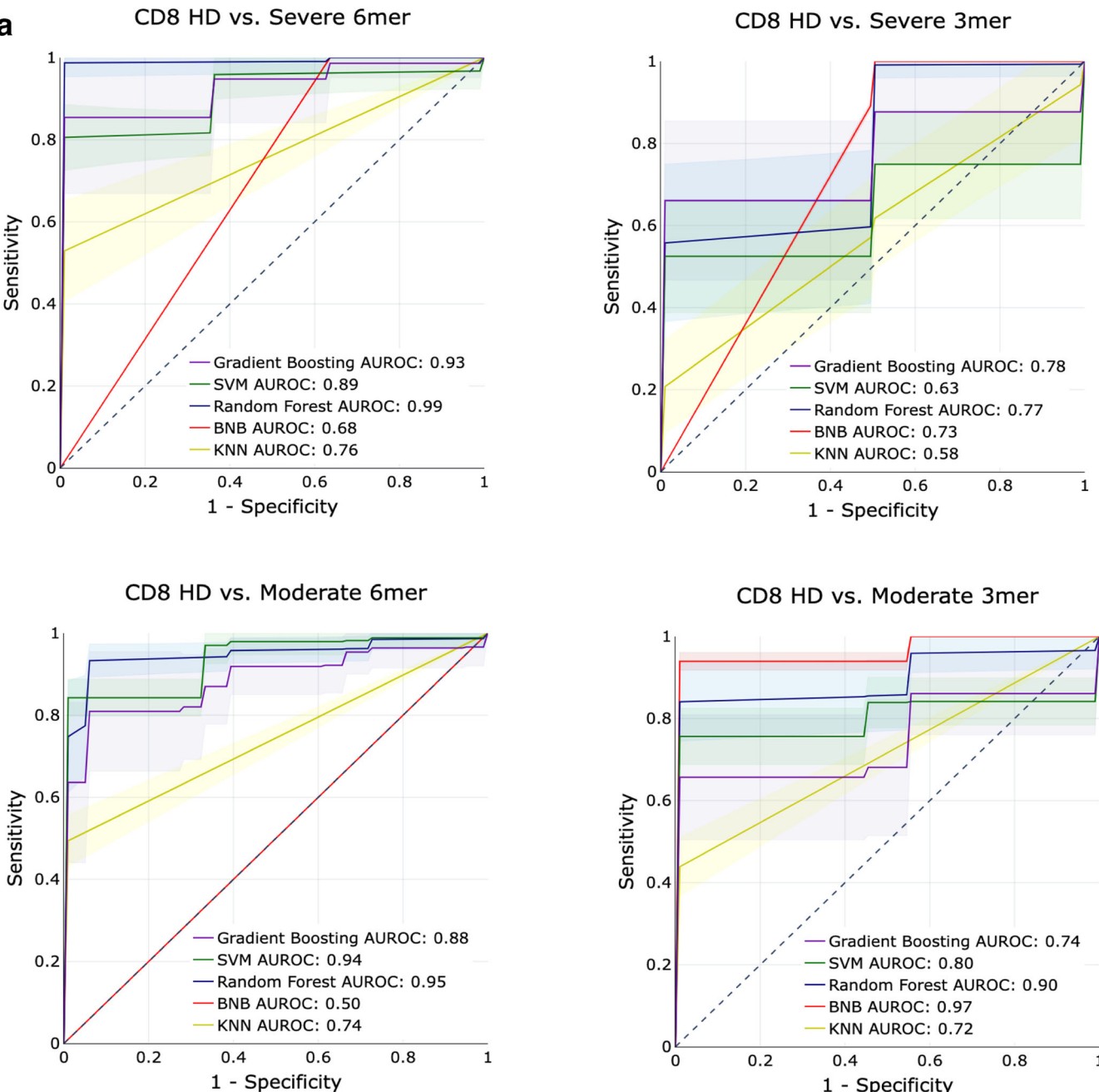

**Fig. 5 Predictive performance of machine learning models for disease severity. a** AUROC curves for five machine learning models (gradient boosting trees, support vector machines, random forests, Bernoulli Naïve Bayes, and k-nearest neighbors) using 6-mer (left) and 3-mer (right) representations of TCR repertoire data. Models were trained to predict disease severity (moderate, severe) vs healthy donors for CD8 samples. Training and evaluation were performed using 500 iterations per model, average performance +/− 1 standard deviation shown on individual plots.

Particularly strong classification performance was observed in the models trained on the CD8 T cell dataset's 6mer representation, with certain predictors approaching near-perfect scores (AUROCs = 0.93–0.99) (Fig. 5A, S7A, Supplementary Data-set S17). Notably, the ML models had a higher performance for classifying moderate COVID repertoires from HD than for classifying severe COVID repertoires from HD. This is consistent with the increased separation of the moderate repertoires observed in the PCA analysis. Model performance in classifying HD from COVID repertoires was much stronger overall in the CD8 T cell subset compared to the CD4 T cell subset, suggesting that immune signatures of COVID infection are much more salient in CD8 T cells' repertoires compared to CD4 T cells'

repertoires (Fig. 5A, Fig. S7A). Overall, these results demonstrate that ML-based methods can identify with high classification accuracy samples from COVID-19 patients of varying severity based on CDR3 sequences features, particularly for moderate disease conditions; however, it should be noted that the performance of these methods have only been demonstrated using the ISB-S datasets and may not be generalizable to other TCR repertoire datasets or for COVID-19 patients more broadly.

## Discussion
T cells are increasingly being recognized as key mediators of viral clearance and host protection in COVID-19, and are subjects of

active investigation[33–35]. However, the rules governing SARS-CoV-2 responsive T cell specificity are still incompletely understood. We provide here a comprehensive, systems immunology approach to analyzing COVID-19 TCR repertoires to discover these rules in an unbiased and systematic manner. By analyzing immune sequencing data from multiple cohorts with TCR-seq data, we found that antigen exposure during the course of COVID-19 decreased the diversity of repertoires and reshaped clonal representation. We identified and characterized enriched CDR3 sequences, k-mer motifs, and patterns associated with disease severity, and found convergent CDR3 gene usages and clusters that have the potential for clonal tracking studies. Comparison of COVID-19-associated motifs and single T cell transcriptomes revealed associations between clonal expansion, disease severity, and cell phenotypes such as effector T cell function. Finally, we established several ML methods for predicting disease phenotypes of varying severities from TCR repertoires, demonstrating high performance for several models and the potential of using ML for prognostication in COVID-19 patients.

Recent studies have started to report on the differences between T cell responses during varying severities of the COVID-19 disease course. Notably, severe COVID-19, albeit having an increase in activated effector cell populations as seen with other disease severities, is associated with lymphopenia and profound functional impairment of CD4 and CD8 T cells[26,36–40]. These results are consistent with our PCA and motif analyses, we observe a stronger signal for the mild and moderate disease repertoires distinguishing them from healthy donors as compared to severe disease repertoires. Moreover, our ML-based methods had higher performance for predicting moderate repertoires, further demonstrating that CDR3 sequence and subsequence features for moderate disease conditions have higher discriminative capacity than for severe conditions. Nevertheless, all of the disease conditions were well differentiated from healthy donors across all analyses suggesting that consensus disease-associated features can be identified, including correlations between clonal expansion in the setting of COVID-19 with effector T cell functions at the transcriptomic level.

With further validation studies, these motifs present in clonally expanded T cells may serve as prognostic and diagnostic biomarkers in COVID infection. Sequence motif studies can be of great clinical significance in broad contexts. Immune cell receptor motif-based investigations are increasingly becoming high-utility, where systematic investigations of specific CDR3 motifs have identified T cell clones associated with specific immune functions like gluten hypersensitivity in celiac disease[41], hyperinflammation in ankylosing spondylitis[42], and reactivity to cancer neoepitopes[43]. As longitudinal studies further uncover physiological and immunological effects of long COVID, where patients experience long-term adverse effects from past COVID infection, it is potentially of great interest to compare the TCR repertoire profiles of these patients to the TCR repertoire features identified here.

Broadly, immune repertoire analysis has become a fundamental tool to understand the biology of immune-mediated diseases as well as immune responses to therapies[44]. T cell receptor motifs have been used as prognostic and diagnostic biomarkers. For example, TCR repertoire studies have shown that improved TCR diversity is linked to better prognostic outcomes for cancer patients who receive immunotherapies[45–47]. This analysis is similar to the analysis we have conducted in our manuscript, where we show the effect of COVID-19 infection on the clonal diversity of T cells. Moreover, TCR repertoire sequencing revealed an increase in hyperexpanded TCR clonal frequencies

following the administration of a neoantigen vaccine, elucidating the role of T cell dynamics in tumor immunology[47]. In light of these findings, similar TCR studies in the context of COVID-19, with a particular emphasis on convalescent patients upon treatment with investigational COVID therapies are of potential interest, to characterize post-treatment T cell dynamics.

Some limitations of this study, however, include the imbalanced nature of the TCR datasets included in the data analyses, the differences in the sizes of datasets that are represented, and the bioinformatics analyses that were carried out independently on each dataset, as opposed to conducting pooled analyses as typical in a meta-analysis. We approach the issue of imbalanced data in our machine learning models by random resampling, specifically by randomly oversampling the minority class in the training dataset. However, limitations to this approach remain, as random oversampling is a naïve technique for rebalancing the class distribution and can result in overfitting in some models. Additionally, the diversity measurements of the TCRs in COVID patients and healthy donors (Fig. 1B, C) are influenced by the sizes of the TCR repertoires in each dataset, as the distributions of T cell receptor frequencies are not linearly scalable. Finally, while a pooled integrative analysis approach is preferable in a meta-analysis to show the generalizability of findings across datasets, this approach was impractical for our study due to the lack of baseline comparability including variations in sequencing modalities, patient populations, and sample sizes. Furthermore, due to differences in data collection protocols among the four datasets, a pooled analysis approach would have likely introduced batch effects that obscure the true biological signal. Instead, separate bioinformatics analysis pipelines were conducted on individual datasets, which allowed all analyses to remain free of batch effects. Despite these limitations, this study offers several insights about the T cell response to COVID-19 infection and employs systems-level approaches to interrogate immunological big data.

To our knowledge, this is the largest scale investigation into TCR specificity groups for COVID-19 to date, spanning 4,730,447,888 clones across 2130 repertoires. Though many studies have sought to identify factors predictive of COVID-19 clinical course and outcomes[48], few have leveraged TCR-seq data and adaptive immune profiles to their full capacity. We provide high-confidence convergent COVID-19-associated signatures with potential prognostic value, including the successful implementation of machine learning models for predicting disease severity. In addition, the use of next-generation sequencing of immune repertoires provides a deeper and more quantitative understanding of the adaptive immune response to COVID-19 and may guide patient risk stratification, vaccine design, and improved clinical management.

## Methods

**Sequence data collection**. TCR repertoire data was obtained from datasets published by Adaptive Biotechnologies[49], ISB-Swedish COVID-19 Biobanking Unit[25], Fifth Medical Center of PLA General Hospital[20], and Wuhan Hankou Hospital China[19]. For COVID-19 patients sequenced with Adaptive Biotechnologies immunoSEQ assays, TCR-seq data were obtained from the ImmuneCODE database at https://doi.org/10.21417/ADPT2020COVID; for healthy donor patients, TCR-seq data was obtained at https://doi.org/10.21417/ADPT2020V4CD. Single-cell TCR-seq and gene expression (GEX) data for CD4+ and CD8+ T cell repertoires from COVID-19 patients and healthy donors from the ISB-Swedish COVID-19 Biobanking Unit[25] was obtained from the ArrayExpress database[50] (http://www.ebi.ac.uk/arrayexpress) using the accession number E-MTAB-9357. Single-cell TCR-seq data from COVID-19 patients and healthy donors were also obtained from the Fifth Medical Center of PLA General Hospital and healthy donors were also obtained from the Fifth Medical Center of PLA General Hospital, accessed through the supplementary tables of the associated publication[20]; and Wuhan Hankou Hospital China, metadata accessed through the supplementary tables of the associated publication[19] and TCR-seq data obtained from the iReceptor platform[51] (http://ireceptor.irmacs.sfu.ca). The Adaptive Biotechnologies dataset comprised bulk TCR-seq data, while the ISB-S, PLAGH, and WHH datasets comprised single-

cell TCR-seq data, with variations in sequencing modalities, patient populations, and sample sizes. Due to the differences in wet lab protocols and the potential presence of batch effects in each of these four datasets, all downstream analyses were performed separately on each individual dataset, and the result from each dataset as well as the consensus findings are reported.

**Data pre-processing**. All four TCR-seq datasets were individually but identically pre-processed for standardized analysis with Immunarch v0.6.6 without pooling[52]. Data obtained from the Adaptive Biotechnologies ImmuneCODE database were used directly as inputs for Immunarch processing, with 1475 COVID-19 patient samples and 88 healthy donor patient samples (1563 samples total) successfully loaded and used for further analysis. For the ISB-Swedish cohort, patients were first filtered by those who had sequencing data available as performed by 10X Genomics. Sequence filtering and processing were performed as follows: for cells with multiple TRA and TRB CDR3 sequences, the first instances, respectively, were selected; only cells with paired TRA and TRB sequences were kept (column chain_pairing = Single pair, Extra alpha, Extra beta or Two chains); sequence files were converted to VDJtools format for input into Immunarch. COVID severity scores were translated from the WHO Ordinal Scale (0–7) to four tiers: healthy donor (0), mild (1–2), moderate (3–4), and severe (5–7). After pre-processing, the CD4 and CD8 datasets were composed of 136,429 and 69,687 clones, represented in a total of 16 healthy donors, 61 mild, 42 moderate, and 24 severe patients, 143 individuals total (16 healthy donors, 108 mild, 93 moderate, and 49 severe repertoires when accounting for patients with samples from two time points, 266 samples total). For the PLA General Hospital and Wuhan Hankou Hospital China cohort, cells with more than one TRA or TRB sequence had the chain with the highest number of reads kept for further analysis, and sequence files were converted to VDJtools format for input into Immunarch. The PLA General Hospital aggregated patient dataset contained 31951 clones across 3 healthy donors (two healthy donors from the original study were excluded for lack of TCR CDR3 amino acid data), 7 moderate, 4 severe, and 6 convalescent patients (of which 4 were the second time point collections of moderate patients—P01, P02, P03, and P04). The Wuhan Hankou Hospital China aggregated patient dataset contained 42001 clones across 5 healthy donors, 5 moderate, and 5 severe patients. Metadata was manually reformatted from supplementary tables.

**Immune repertoire statistics**. Clonotype statistics and diversity metrics were calculated using Immunarch v0.6.6[52]. For the total number of unique clonotypes, the repExplore function was used with parameter.method = volume; for distribution of CDR3 sequence lengths, repExplore function with.method = len and.col = aa; for the Chao1 estimator, repDiversity function with.method = chao1; for Gini-Simpson index, repDiversity function with.method = gini.simp; for Inverse Simpson index, repDiversity function with.method = inv.simp. Clonal proportion estimates were calculated with the repClonality function with.method = top. CDR3, V gene, and J gene usage proportions were calculated and aggregated directly from sample TCR data. Statistical significance testing comparing groups was performed using the two-sided Wilcoxon rank-sum test by the wilcox.test in R.

**K-mer analyses**. For K-mer abundance calculations, each VDJtools formatted sample was converted to a vector of CDR3 sequences. The vector was converted to k-mer statistics using the getKmers function from Immunarch, then merged with k-mer statistics of other samples using the R function merge with parameter all = TRUE for full outer join. Empty cells were converted from NAs to 0 counts. The 50,000 top variance unique k-mers were selected for downstream analyses (PCA and machine learning pipelines) with the exception of 3-mers which had 6916 unique k-mers. The selection of 50,000 top variance k-mers was done to keep the data dimensions consistent, increase the efficiency of the analysis pipelines, and use data features that are the most likely to be the most biologically meaningful. Low-variance k-mers, either due to lack of representation or due to conservation between healthy and disease samples are unlikely to play an important role in the T cell response to COVID given that the signature is not shared or enriched by disease status. K-mer counts were normalized to sum to 1 for each sample prior to downstream analyses. PCA was performed using the prcomp function in R with parameter center = TRUE. All data shown in the PCA plots (Fig. 2A, C, S4A–F) were generated from only the ISB-S dataset, all of whose samples were sequenced under a uniform protocol, which ensured that clustering and separation in the dimensionality reduction were attributable to intrinsic differences rather than batch effects.

**Motif analyses**. TCR clustering and specificity group analysis was performed using GLIPH2[29]. Software executable for analysis was obtained from http://50.255.35.37:8080/ and run with the human v2.0 reference on clonal data for each disease condition and T cell type. Parameters include global_convergence_cutoff = 1, local_min_OVE = 10, kmer_min_depth = 3, simulation_depth = 1000, p_depth = 1000, ignored_end_length = 3, cdr3_length_cutoff = 8, motif_distance_cutoff = 3, all_aa_interchangeable = 1, kmer_sizes = 2,3,4, and local_min_pvalue = 0.001000.

Generation probability calculations were performed using OLGA[30]. Software installation and setup were performed as described in https://github.com/statbiophys/OLGA and run on clonal data for each disease condition and T cell type. Representative calculations with parameters are as follows: olga-compute_pgen -i input.tsv –humanTRB -o out_pgens.tsv –v_in 1 –j_in 2.

**Single-cell transcriptome analyses**. Single-cell transcriptome data from the ISB-S dataset were processed using Seurat v4.0.4. The pipeline included log normalization with a scale factor of 1,000,000, scaling and centering, PCA, nearest-neighbor graph construction, clustering with the Louvain algorithm, UMAP, differential gene expression, and generation of various visualizations. Parameters included: for the FindNeighbors function, dims = 1:10; for FindClusters, resolution = 0.6; for RunUMAP, dims = 1:10; for FindAllMarkers, only.pos = TRUE, min.pct = 0.25, logfc.threshold = 0.25. Differential gene expression between clonally expanded clusters and all other cells was performed using a downsampled cell subset (5000 cells per group) of the data and the FindMarkers function with parameters logfc.threshold = 0.01 and min.pct = 0.1. P-value adjustment was performed using Bonferroni correction. Upregulated or downregulated genes with significance q-value < 1e-4 were then used for functional annotation with DAVID analysis. In addition to default Seurat outputs, custom R scripts were used to generate visualizations including UMAPs associated with CDR3 motifs and disease severity.

**Training and evaluation of k-mer-based machine learning models**. Five ML-based approaches were trained on the k-mer frequency matrix generated from amino acids in the CDR3 region in the T cell repertoires of healthy donors and COVID-19 patients from the ISB-S datasets, using Python v3.8.6 and scikit-learn v0.23.1. These algorithms were: Random Forests (RF), Support Vector Machines (SVM), Bernoulli Naïve Bayes (BNB), Gradient Boosting Classifiers (GBC), and K-Nearest Neighbors (KNN). The k-mer frequency matrix dataset was partitioned into subsets to perform binary classification between the healthy donor and the specified disease phenotype, such that models were trained for classification tasks of healthy donor vs moderate disease and healthy donor vs severe disease. The dataset was first partitioned into test and train sets with an 80:20 ratio. Following this test-train partition, to address imbalanced data, healthy donor samples were randomly resampled to be equal to the number of COVID-19 samples represented in the dataset, prior to training. Hyperparameter selection was informed by GridSearchCV, where optimal parameters found by the grid search were adopted when empirical performance on the test set was improved from the default parameters. For the CD8 subset, RFs were trained with 100 estimators, the Gini impurity criterion for measuring the quality of splits, minimum samples required to split an internal node of 2, minimum number of samples required to be a leaf node of 1, and bootstrapping to build trees. For the CD4 subset, RFs were trained with 2000 estimators, Gini impurity criterion for measuring the quality of splits, minimum samples required to split an internal node of 5, and bootstrapping to build trees. For the CD8 subset, SVMs were trained with the polynomial kernel, parameters C = 20, degree = 5. For the CD4 subset, the SVMs were trained with the RBF kernel, C = 100 and gamma = 1. For the CD8 subset, NBs were trained with alpha = 1.0, binarize = 0.0, fit_prior = True, and Class_prior = None. For the CD4 subset, NBs were trained with alpha = 1e-8 (approximating zero), binarize = 0.0, fit_prior = True, class_prior = [0.5, 0.5]. For CD8, GBCs were trained with 100 estimators, a learning rate of 1.0, and a maximum depth of 1. For CD4, GBCs were trained with 100 estimators, a learning rate of 1.0, and a maximum depth of 20. For CD8, KNNs were trained with $k = 3$, leaf size of 30 and the Minkowski distance metric. For CD4, KNNs were trained with $k = 3$, leaf size of 10 and the Minkowski distance metric. For five iterations per repeat and for 100 repeats (a total of 500 model evaluations), estimators were trained on a random 80% of the previously partitioned train set and subsequently evaluated on the test set. Plotly v5.1.0 was used to generate ROC plots from performance results.

**Statistics and reproducibility**. Comprehensive information on the statistical analyses used is included in various places, including the figures, figure legends and results, where the methods, significance, p-values, and/or tails are described. All error bars have been defined in the figure legends or methods. Standard statistical calculations such as Spearman's rho were performed in R with functions such as cor.

**Graphical illustrations**. Certain graphical illustrations were made with BioRender (biorender.com).

**Reporting summary**. Further information on research design is available in the Nature Portfolio Reporting Summary linked to this article.

## Data availability

The authors are committed to freely sharing all COVID-19-related data, knowledge, and resources with the community to facilitate the development of new treatment or prevention approaches against SARS-CoV-2/COVID-19 as soon as possible. All relevant processed data generated during this study are included in this article and its

supplementary information files, or have been deposited to Figshare at: https://figshare.com/projects/COVID-TCR_Open_Data_Deposit/154308. Raw data are from various sources as described above. Any additional data and resources related to this study are freely available upon request to the corresponding author.

## Code availability

Key code used for data analysis or generation of the figures related to this study have been included in this article and its supplementary information files, and have been deposited to Zenodo at https://doi.org/10.5281/zenodo.7359175. Additional scripts used are also available upon request to the corresponding author.

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

## Acknowledgements

We thank various members from Chen lab for informative discussions. We thank staff members from Yale HPC for technical support. We thank various support from the Department of Genetics; Institutes of Systems Biology and Cancer Biology; Dean's Office of Yale School of Medicine and the Office of Vice Provost for Research. This work is supported by DoD PRMRP IIAR (W81XWH-21-1-0019) and discretionary funds to SC.

## Author contributions

JJP and SC conceived and designed the study. JJP developed the analysis approach, performed data analyses, and created the figures. KAVL performed data analyses and established machine learning pipelines. SZL performed pre-processing of datasets. JJP, KAVL, and SC prepared the manuscript with input from SZL, KM, and ZF. SC supervised the work.

## Competing interests

The authors declare no competing interests.
