## [Peer Review File · Communications Biology]

Reviewers' comments:

Reviewer #1 (Remarks to the Author):

This is an interesting and potentially highly clinically relevant study, using data from patient blood samples to try to determine signatures associated with COVID-19 disease severity, primarily using machine learning models for classification on the basis of TCR repertoire analysis and then combining such data/ analysis with transcriptomic data from single cell studies. Figures are well presented and the bioinformatic methods are interesting.

Key concerns are (1) a lack of consideration of bias due to different laboratory sequencing methodologies for the different cohorts, which have been drawn together, and (2) the extent to which the same samples were used in both training and testing sets, leading to an overestimate of the performance of the machine learning methods.

Specific major comments:

Page 3, lines 29-32: Introduction - I am unsure of the reason for discussing a largely irrelevant, clonal process (lymphoblastic leukaemia), while not properly reviewing the literature in terms of single cell or bulk TCR repertoire sequencing work that has contributed to our understanding of SARS-CoV-2 responses.

Page 4, lines 19-24: Results – This is a key methodological concern: A series of datasets appear to be “lumped together” without any consideration of potentially differing laboratory methods, whether CD4/CD8 or all T-cells were sequenced or the amount of starting material (i.e., what numbers of lymphocytes the nucleic acid sample used in sequencing equated to). No consideration appears to be given to potential bias introduced by any of this. The manuscript is very much written from a bioinformatician’s standpoint, without critically considering the “wet lab” methods. If I have misunderstood this, I apologise, but it is not at all clear from the manuscript what has been compared with what at various points in the analyses.

Page 9, lines 15-17 again demonstrate the mindset on page 4, considering only bioinformatic analysis: “By uniformly processing immune sequencing data from multiple cohorts with TCR-seq data, we found that antigen exposure during the course of COVID-19 significantly decreased the diversity of repertoires and reshaped clonal representation.” Uniform bioinformatic processing cannot negate the effects of using potentially mismatched cohorts produced using different laboratory methodology with different biases (please see my comment above). I have not had time to go through the various cohorts, but the authors need to reassure the reader that they are comparing like with like, rather than “apples with oranges”, so to speak.

Page 4, line 5: “fewer differences were found for the ISB-S CD4 and CD8 datasets when comparing samples from different disease severities to those from healthy donors”. Following from the two concerns above, did the healthy donors’ samples contain sequences from both CD4 and CD8 T-cells? This will dilute any signal obtained.

Page 4, lines 7-8: “By comparison, the top CDR3 sequences were different across conditions for both the AB and ISB-S datasets.” Again, is this due to differences between laboratory methodology or a genuine difference between patients? This does not appear to be considered.

Page 13, lines 28-29: It is very important that the same upsampled data is not present in both the training and testing splits, because in that case the authors would be testing on the training data and artificially increasing the cross-validation performance. The authors should provide reassurance that this isn’t happening. However, from inspection of the code, it appears that they are testing on the

training data. This is not an appropriate way to assess a machine learning method's performance.

Page 9 lines 2-3: It would be helpful for clarity if results were specifically attributed to the testing data partitions here, if that is the case.

Page 9 lines 6-9: It appears that separate models for CD4 and CD8 analysis are used (which is helpful, in light of the above comments), and therefore the authors cannot use the Adaptive dataset for hold-out testing. However, it would be good practice if they acknowledged that their cross-validation does not necessarily indicate good generalisation of their machine learning models.

Page 10 lines 9-10: Given that no hold-out testing was carried out, the claim of successful prediction of disease severity cannot be substantiated and should be removed or made substantially more speculative.

Page 14 line 1: It would be good to know what the value of K is in the K-nearest neighbour classifier. Maybe it was set using the defaults. It would be helpful to mention whether default parameters are used in any other methods too, as they may not be appropriate for the analysis being done here.

Page 4 lines 9-10: It might be helpful to explain why different thresholds are used for different datasets.

Specific minor comments:

Page 7 lines 11-13: This sentence doesn't make sense to me: "Comparison with the top enriched motifs found from the GLIPH2 analysis, including AGQGA%E, S%AAG, SL%AG, SLQGA%YE, S%SGTDT, SL%GTDT, SLS%TDT, and S%AGNQP revealed high density of clusters in cluster 6"

Page 11 lines 21-22: This sentence doesn't make sense to me (either semantically or grammatically): "For the ISB-Swedish cohort, patients were first filtered by those were sequenced by 10X Genomics."

Page 13 line 32: Used acronym SVC instead of SVM. I think support vector machines have been mixed up with the name of the sklearn function, support vector classification, which may be confusing to the reader.

Reviewer #2 (Remarks to the Author):

This article describes an integrative and immunology approach to systematically study more than 4 million TCR sequences from different sources for COVID-19 patients and healthy donors. Overall, this is a valuable effort that helps revealed patterns of the adaptive immune response during SARS-CoV-2 infection. In addition, through the analysis of biological pathways, the authors also suggested that T cell clonal expansion is highly related to some T cell effector functions and TCR signaling. However, some issues or concerns need to be addressed.

1. For k-mers analysis, why did the author set 50,000 top variance unique k-mers for downstream analyses e.g., PCA and machine learning pipelines. Will different threshold settings bring different clustering and performance?
2. In addition to K-mers, did the author consider using other methods, such as deep learning, multiple sequence alignments and other methods for clustering and other downstream analysis?
3. In the machine learning part, because the number of healthy donor samples is relatively small, even though the author uses the unsampled method, is there any possibility of overfitting? Because I have observed that the performance of many classifiers is 1. I suggest that the model performance can be re-evaluated on the independent test set collected separately.
4. Regarding the predictor, the author did not release an independent program or online predictor, so

it is not available for others to test or use.

5. Another key question is whether there is any clinical application of the findings in the author's article. What is the point of finding these important TCR patterns? Such as the development of personalized vaccines, observation of potential COVID-19 patients.

6. The author constructed 3-mer, 4-mer, 5-mer, and 6-mer frequency matrix representations of ISB-S CD4 and CD8 datasets and performed PCA analysis to see whether samples cluster by disease severity. I observed that the majority of samples clustered together, although a number of mild and moderate samples were separated from the main cluster. So, is it possible that this part of the separated samples is caused by sequencing technology or batch effects? I think the conclusion the author got here is not supported by statistical significance.

Reviewer #3 (Remarks to the Author):

This paper is a meta-analysis of TCR data from COVID-19 and healthy controls. The authors bring together data from a number of previous studies, and provide the single biggest set of TCR data from COVID-19 patients. The paper makes a number of claims about the COVID-19 repertoires. As in all such metadata, especially where the data were produced by completely different methodologies, it is absolutely crucial to provide full clarity about the composition of the different data sets. One key element is to show how many of each cohort were mild, moderate or severe, or HD. I may have missed this somewhere in the supplementaries, but it is essential to make this data obvious in fig 1. I suspect the cohorts are very unbalanced.

In fig 1, all the estimators are strongly influenced by sample size. The authors should show their conclusions are not influenced by sample size. I could not understand what panel E represented. In F, was this just a random sample of 32 AB repertoires? Why 32? Was this repeated multiple times?

In Fig 2, the PCAs are not convincing to me. There are a few outlying mild and moderate, but otherwise they look totally overlapping.. In the heatmaps, are these averages of multiple subsamples? Is there any obvious separation between condition if the samples are allowed to cluster across the different individuals? I did not really understand the import of the panels G and H. Could the authors explain this a bit more clearly?

In Fig 3, what is the proportion of cells between different disease subsets? Can the authors say a bit more about the subsets? Which are naïve? Is there a proliferating subset? The authors say there is a relationship between the cluster pattern and disease severity – but it looks more like a radically different distribution between healthy and COVID-19 repertoires. This seems a striking finding, and not necessarily what other people have observed? This figure deserves a much more detailed discussion and analysis.

Fig 4 is lacking a lot of detail which is important for interpretation. How balanced are the repertoires? What is meant by upsampling the controls? How were the parameters decided – was this done on an independent data set (i.e. before the cross-validation)? Did the authors try fitting models on one data set, and then testing on one of the other data sets?

Overall, the attempt to provide a meta analysis of diverse TCRrep data sets is interesting and potentially valuable. But in light of the known challenges of meta-analyses, the paper is a bit skimpy in providing detail which could be used to assess the validity of the comparisons. Also, at the end of the day, each analysis has largely been carried out on an independent data set, which rather reduces the value of carrying out a meta-analysis in the first place. Other than the overall reduction in diversity shown in fig 1, I am not convinced any of the other findings are generalizable across data sets, and so its hard to assess the significance of the findings.

Response letter for revision of Park et al., “T cell receptor repertoire signatures associated with COVID-19 severity”

Summary of this revision

- Completed all feasible suggested / requested analyses.
- Clarified the rationale for bioinformatics processing pipelines and analysis methods of choice.
- Provided new machine learning models that demonstrate enhanced generalizability using motif-based data representation approach
- Revised the figures regarding machine-learning performance to new results from rigorous hold-out testing.
- Provided requested code in a public Github repository.
- Clarified the details of the methodology.
- Addressed all remaining points in a point-by-point manner.

Details to follow below.

Point-by-point response to the reviewers (Park et al., *Communications Biology* manuscript)

Reviewer #1: This is an interesting and potentially highly clinically relevant study, using data from patient blood samples to try to determine signatures associated with COVID-19 disease severity, primarily using machine learning models for classification on the basis of TCR repertoire analysis and then combining such data/ analysis with transcriptomic data from single cell studies. Figures are well presented and the bioinformatic methods are interesting.

Key concerns are (1) a lack of consideration of bias due to different laboratory sequencing methodologies for the different cohorts, which have been drawn together, and (2) the extent to which the same samples were used in both training and testing sets, leading to an overestimate of the performance of the machine learning methods.

Response:

We thank the reviewer for an overall accurate summary and positive comments on the importance and novelty of the project. We agree with the reviewer that consideration of different laboratory sequencing methodologies for different cohorts is critical¹. Batch effect correction, as the reviewers point out, is particularly critical for pooled analyses of multiple datasets to ensure biological comparability and the discernment of meaningful biological signal from sequencing noise. However, as we will clarify in our subsequent responses, each of our analysis pipelines were run separately on the individual datasets, rather than being pooled and analyzed together. The separate generation of summary statistics on each dataset, albeit using the same computational tools for their generation, highlights that no “merging” or “pooled” analyses were conducted. As such, batch effect correction procedures are not critical issues in our manuscript.

We acknowledge that with very specific dataset selection criteria, it may be possible to integrate immune repertoire datasets together with computational batch effect correction and subsequently analyze in a pooled approach. Notably, this approach may be feasible if all datasets are single cell immune repertoire sequencing data and are also meaningfully comparable in terms of the cell types included and laboratory methods employed. Unfortunately, the datasets in this manuscript includes a bulk-TCR sequencing dataset (Adaptive Biotechnologies), which cannot be readily integrated with the single-cell datasets (ISB-S, PLA-H, WHH).

It is, in theory, possible to perform integrative and pooled analyses for the three single-cell datasets if we were primarily interested in gene expression via RNA-seq. In single-cell RNA-seq, powerful computational tools such as Seurat², Harmony³, Mutual Nearest Neighbors⁴, and LIGER⁵, have enabled

¹ Tung, P.Y., Blischak, J., Hsiao, C. *et al.* Batch effects and the effective design of single-cell gene expression studies. *Sci Rep* 7, 39921 (2017). <https://doi.org/10.1038/srep39921>

² Stuart, T., Satija, R., *et al.* 2019. Comprehensive Integration of Single-Cell Data. *Cell*, 177(7), pp.1888-1902.e21.

³ Korsunsky, I., Millard, N., Fan, J. *et al.* Fast, sensitive and accurate integration of single-cell data with Harmony. *Nat Methods* 16, 1289–1296 (2019). <https://doi.org/10.1038/s41592-019-0619-0>

⁴ Haghverdi, Laleh, *et al.* Batch effects in single-cell RNA-sequencing data are corrected by matching mutual nearest neighbors. *Nature biotechnology* 36.5 (2018): 421-427.

⁵ Welch, Joshua D., *et al.* Single-cell multi-omic integration compares and contrasts features of brain cell identity. *Cell* 177.7 (2019): 1873-1887.

batch-effect corrected integration of multiple datasets. However, our analysis is focused on TCR-seq data from the multiple datasets, which has unique challenges compared to RNA-seq data when considering the possibility of batch effect correction. While scRNA-seq quantifies expression levels of genes, where intuitively, some baseline signal of the “batch” from each dataset can be subtracted from the background to separate “signal” from “noise,” in scTCR-seq, the data structure is comprised of sequence information rather than quantifiable expression data. We therefore focused our analyses on gaining insights into changes in repertoire diversity, specificity, and clonal composition within particular datasets.

Additional considerations that reinforced our decision to pursue our analytical approach were that the success of computational approaches for batch effect correction is most salient among datasets with high baseline similarity. Single-cell RNA-seq analyses that have seen the most success in employing these computational tools for multi-dataset integration were mostly integrating together highly comparable individual datasets. For example, the commonly used multi-dataset integration tool in Seurat offers a tutorial where integration is performed on two datasets generated by the same lab, using identical sequencing methods⁶. The datasets in our manuscript lack such a high degree of baseline similarity. They come from different laboratories located in multiple nations, which, as the reviewers point out, differed in wet-lab techniques and sequencing protocol. Crucially, the ISB-S dataset used droplet-based sequencing for the TCR sequences, whereas the PLA and Wuhan Hankou dataset used full-length sequencing, which reduces the baseline comparability of the datasets in the first place. Also, the three single-cell datasets are not of equal size. The ISB-S (Su) dataset vastly outnumbers both the WHH (Wen) and PLA Hospital (Zhang) datasets in sample size. If a pooled analysis approach is undertaken, we anticipate that the biological signal from the ISB-S dataset will be disproportionately amplified in the results, compared to the biological signal from the PLA and WHH datasets. As such, we believe that an analytical pipeline that separately analyzes the individual datasets and reports the results from each dataset as well as the consensus results, which is the approach in our manuscript, is more methodologically sound than attempting a pooled approach.

Our initial machine learning analyses submitted in the initial manuscript (Figure 5, Supplemental Figure 7) were also conducted only on the ISB datasets, and not on any of the other datasets. We later expanded on our machine learning analyses to the other datasets included in our manuscript to demonstrate the generalizability of our models, and we show that the performance of our machine learning models are robust even in cross-dataset analyses. Thus, we hope to clarify the scope of our analyses and demonstrate why some of the key overarching concerns regarding batch effects do not affect our analyses per reviewer comments.

Specific major comments:

Page 3, lines 29-32: Introduction - I am unsure of the reason for discussing a largely irrelevant, clonal process (lymphoblastic leukaemia), while not properly reviewing the literature in terms of single cell or bulk TCR repertoire sequencing work that has contributed to our understanding of SARS-CoV-2 responses.

⁶ Kang, H., Subramaniam, M., Targ, S. *et al.* Multiplexed droplet single-cell RNA-sequencing using natural genetic variation. *Nat Biotechnol* **36**, 89–94 (2018). <https://doi.org/10.1038/nbt.4042>

We thank the reviewer for this comment and agree that it is important to review the literature of TCR repertoire sequencing that has contributed to our understanding of SARS-CoV-2 responses. We have updated our manuscript to review the literature in terms of TCR sequencing work that has contributed to our understanding of SARS-CoV-2 immunity. In the original submission, the lymphoblastic leukemia example was mentioned to illustrate the broader clinical utility of TCR repertoire sequencing studies for prognostic biomarker development.

Page 4, lines 19-24: Results – This is a key methodological concern: A series of datasets appear to be “lumped together” without any consideration of potentially differing laboratory methods, whether CD4/CD8 or all T-cells were sequenced or the amount of starting material (i.e., what numbers of lymphocytes the nucleic acid sample used in sequencing equated to). No consideration appears to be given to potential bias introduced by any of this. The manuscript is very much written from a bioinformatician’s standpoint, without critically considering the “wet lab” methods. If I have misunderstood this, I apologise, but it is not at all clear from the manuscript what has been compared with what at various points in the analyses. We thank the reviewer for expressing these concerns. However, once again we clarify that no datasets were lumped together for analysis; instead, each dataset was analyzed with a set of bioinformatics tools, and the consensus results of this analysis is being reported.

Page 9, lines 15-17 again demonstrate the mindset on page 4, considering only bioinformatic analysis: “By uniformly processing immune sequencing data from multiple cohorts with TCR-seq data, we found that antigen exposure during the course of COVID-19 significantly decreased the diversity of repertoires and reshaped clonal representation.” Uniform bioinformatic processing cannot negate the effects of using potentially mismatched cohorts produced using different laboratory methodology with different biases (please see my comment above). I have not had time to go through the various cohorts, but the authors need to reassure the reader that they are comparing like with like, rather than “apples with oranges”, so to speak.

We thank the reviewer for this comment. Once again, we wish to clarify that uniform bioinformatic processing entailed separate but standardized processing of individual datasets using identical pipelines (the bioinformatics tools used broadly in the literature), as opposed to combining the data into one pooled set prior to analysis. We believe that this key distinction allows us to ensure comparability of results that are being compared in this manuscript because all comparisons are made within each dataset. When any claims in the paper mention multiple datasets, they are simply reporting consensus results of the within-dataset comparisons. While we agree with the reviewer that these concerns are important to address when multiple datasets are merged together, we believe that these concerns are not applicable in our manuscript.

Page 4, line 5: “fewer differences were found for the ISB-S CD4 and CD8 datasets when comparing samples from different disease severities to those from healthy donors”. Following from the two concerns above, did the healthy donors’ samples contain sequences from both CD4 and CD8 T-cells? This will dilute any signal obtained.

We thank the reviewer for this comment. Yes, the healthy donors’ samples in the ISB dataset contained sequences from both CD4 and CD8 T cells. All comparisons between HD and COVID patients are performed separately in CD4 vs. CD8 T cell datasets in our manuscript. We believe that confusion here may have stemmed from the ambiguity in the language of this sentence on page 4 line 5, and we have

reworded this sentence in the manuscript to enhance clarity. For the ISB-S dataset, the authors of the original manuscript (Su et al) have provided separate data for CD4 and CD8 T cells, including for healthy donors. As such, sequences from healthy donors and COVID come as a set in each of the CD4 and CD8 datasets. Thus, all our comparisons are technically restricted in this manner: in other words, our comparative analyses are partitioned for CD4 (HD vs. COVID) and CD8 (HD vs. COVID). Thus, we hope to reassure the reviewers that no such signal dilution has occurred in our analysis. The sentence here is trying to convey that the difference between the TCR samples of COVID-19 patients and healthy donors from the AB dataset were greater than the differences observed between TCR samples of different COVID severities (e.g. Mild vs. Moderate vs. Severe COVID) from the ISB-S datasets. Therefore, the presence of both CD4 and CD8 T cells in the healthy donor dataset does not dilute any signal observed, as comparisons are made separately among CD4 T cells and among CD8 T cells.

Page 4, lines 7-8: “By comparison, the top CDR3 sequences were different across conditions for both the AB and ISB-S datasets.” Again, is this due to differences between laboratory methodology or a genuine difference between patients? This does not appear to be considered.

We thank the reviewer for this comment. We apologize for the lack of clarity in the original manuscript, as we now see how this sentence could be misinterpreted as a comparison between the two datasets, rather than a comparison of CDR3 representation across the disease severities within each dataset. The wording in the manuscript has been modified to enhance clarity. We wish to clarify that all comparisons between healthy donors and COVID patients take place within datasets, and any claims in the manuscript that refer to multiple datasets are simply reporting the consensus findings of within-dataset comparisons. For example, healthy donors in the Adaptive Biotechnologies (AB) dataset are only compared to COVID patients in the AB dataset. Here, we are not making the claim that the sequences in AB and ISB-S datasets were different, but rather that in each dataset, the CDR3 sequence that are most frequently represented in each level of disease severity are different. We believe this finding is interesting because the distinctness of the top CDR3 sequences found in each disease severity is suggestive of how signatures of severity of COVID-19 infection can be “read” out in the T cell receptor repertoire.

The differences in the top CDR3 sequences represented at each level of COVID-19 severity may be reflective of differences in clonal expansion of relevant T cell subpopulations that may vary in their ability to effectively mount an immune response against the SARS-CoV-2 virus. Given the sheer size of the Adaptive Biotechnologies dataset, which has TCR repertoires collected data from 1563 patients, we are convinced of the robustness of this signal that T cells expressing certain CDR3 sequences are disproportionately clonally expanded in patients at different levels of disease severity. The fact that this observation of differing CDR3 representations by disease class was found in two of our largest datasets, AB and ISB-S, further adds significance to this observed trend.

Page 13, lines 28-29: It is very important that the same upsampled data is not present in both the training and testing splits, because in that case the authors would be testing on the training data and artificially increasing the cross-validation performance. The authors should provide reassurance that this isn’t happening. However, from inspection of the code, it appears that they are testing on the training data. This is not an appropriate way to assess a machine learning method’s performance.

We thank the reviewer for these comments. We agree that the upsampling of the healthy donor and COVID-19 sequences prior to training the machine learning algorithms makes it imperative that the same

upsampled data is not present in both training and test sets. As this is a crucial step in ensuring the integrity of our analyses, we have significantly modified our code to ensure that the data in the train and test sets are distinct. We found that the models in the original manuscript had these issues. Nonetheless, after correcting for this methodological issue, we still attain high model performance, particularly in the CD8 subset where several algorithms attain AUROCs > 0.90 , which is consistent with the findings in the original manuscript.

Page 9 lines 2-3: It would be helpful for clarity if results were specifically attributed to the testing data partitions here, if that is the case.

The results described are attributed to the classification of a given patient's TCR repertoire kmer profile as either HD or COVID-19. "Each permutation" refers to the length of the kmers ($k = 3, k = 6$) as well as the classifier's training for classifying HD vs. each level of severity of COVID TCR profiles. We have updated the manuscript to enhance the clarity as to what the permutations refer.

Page 9 lines 6-9: It appears that separate models for CD4 and CD8 analysis are used (which is helpful, in light of the above comments), and therefore the authors cannot use the Adaptive dataset for hold-out testing. However, it would be good practice if they acknowledged that their cross-validation does not necessarily indicate good generalisation of their machine learning models.

Page 10 lines 9-10: Given that no hold-out testing was carried out, the claim of successful prediction of disease severity cannot be substantiated and should be removed or made substantially more speculative.

We thank the reviewer for these comments and suggestions. We agree with the reviewer that the Adaptive Biotech dataset is not suitable for hold-out testing. We do not believe it would be appropriate to train models on a single cell TCR dataset and then subsequently do hold-out testing on a bulk TCR dataset. Such an analysis is neither feasible nor appropriate because they are not comparable data at baseline. As the reviewer points out, the separation between CD4 vs. CD8 cell types is also another feature of the AB dataset that differs from the ISB dataset, on which our machine learning models were trained.

Still, we agree with the reviewer that using one dataset for training machine learning models and testing their performance on another dataset will provide greater assurance of the generalizability of the machine learning models. Per the reviewer's suggestions, we have attempted several different methods for hold-out testing of our machine learning models. We first tried training our machine learning models on ISB-S's 6mer matrices and subsequently testing their performance on PLA and WHH datasets' 6mer matrices. Unfortunately, these models did not attain high performance because very few 6mers that were represented in the ISB-S patients' TCR repertoires were also represented in the PLA and Wuhan Hankou patients' TCR repertoires.

While our kmer models were not generalizable to other datasets, we successfully implemented an alternative data curation approach using motif-based clustering, which enabled the generalizability of our machine learning models to external validation sets. While the different datasets were unlikely to include identical kmers, we hypothesized that CDR3 motifs would still be found in common across datasets, so we opted to curate our data as a motif-based counts matrix using GLIPH2 motifs instead of kmers. With this new data curation approach, we were able to train our machine learning models on the ISB-S dataset and assess performance on the PLA and WHH datasets as an external hold-out testing validation. We trained the models on two permutations (HD vs. Moderate and HD vs. Severe) on the ISB-S dataset and

tested the models on the equivalent subsets of the PLA (Zhang) and WHH (Wen) datasets. As the PLA and WHH datasets lack any mild COVID patients, we did not try this permutation. We also built soft-voter ensemble classifiers of the five classifiers (Random Forest, Support Vector Machine, Gradient Boosting, KNN and Bernoulli Naïve Bayes), which generated highly accurate predictions in the validation sets. The results are reported below.

GLIPH2 Models: HD vs. Severe, Train = Su, Test = Wen

GLIPH2 Models: HD vs. Severe, Train = Su, Test = Zhang

GLIPH2 Models: HD vs. Moderate, Train = Su, Test = Wen

GLIPH2 Models: HD vs. Moderate, Train = Su, Test = Zhang

**ROC Curves of GLIPH2-based models, Five Supervised ML Classifiers:
External Validation Set Holdout Testing Results –
Training on ISB-S, Testing on PLA (Zhang) and WHH (Wen)**

Soft Voter Ensemble: Test = Wen Severe (AUC=0.72)

Soft Voter Ensemble: Test = Zhang Severe (AUC=1.00)

Soft Voter Ensemble: Test = Wen Moderate (AUC=0.84)

Soft Voter Ensemble: Test = Zhang Moderate (AUC=0.71)

**ROC Curves of GLIPH2-based models, Soft Voter Ensemble Classifiers:
External Validation Set Holdout Testing Results –
Training on ISB-S, Testing on PLA(Zhang) and WHH (Wen)**

Page 14 line 1: It would be good to know what the value of K is in the K-nearest neighbour classifier. Maybe it was set using the defaults. It would be helpful to mention whether default parameters are used in any other methods too, as they may not be appropriate for the analysis being done here.

We thank the reviewer for these suggestions. The value of K in the K-nearest neighbor classifier is $k = 3$ which we have now clarified in the methods section. While default parameters attained high classification performance, nevertheless, we agree that it is important to explain the rationale for hyperparameter selection. In response to these comments, we have attempted hyperparameter optimization of our models via a grid search of candidate parameters using the GridSearchCV function in scikit-learn's model_selection library. The Grid Search was performed on a 6mer dataset that combines CD4 and CD8 TCR repertoires of the ISB-S dataset to classify HD from all COVID sequences, so that the same hyperparameters could be used to train on CD4 and CD8 TCR data for all permutations (HD vs. Severe, HD vs. Moderate, HD vs. Mild).

Algorithm	Hyperparameters in Grid Search	Optimal Hyperparameters according to GridSearchCV
SVM	<pre>param_grid = { 'C': [0.1,1, 10, 100], 'gamma': [1,0.1,0.01,0.001], 'kernel': ['rbf', 'poly', 'sigmoid', 'linear'] } </pre> Fitted 5 folds for each of 64 candidates, totaling 320 fits.	C = 100, Gamma = 1, Kernel = RBF
KNN	<pre>param_grid = { 'n_neighbors': [3,4,5,6,7,8,9,10], 'weights': ["uniform","distance"], 'algorithm': ["auto","ball_tree","kd_tree","brute"], 'leaf_size': [10,20,30], 'p': [1, 2] } </pre> Fitted 3 folds for each of 384 candidates, totaling 1152 fits.	K = 3, Weights = Distance, Algorithm = Auto, Leaf Size = 10, p = 2 (Distance Type is Euclidean)
Random Forest	<pre>param_grid = { 'n_estimators': [200, 400, 600, 800, 1000, 1200, 1400, 1600, 1800, 2000], 'max_features': ['auto', 'sqrt'], 'max_depth': [10, 20, 30, 40, 50, 60, 70, 80, 90, 100, 110], 'min_samples_split': [2, 5, 10], 'bootstrap': [True, False] } </pre> Fitted 3 folds for each of 1440 candidates, totaling 4320 fits.	Bootstrap = True, Max_Depth = 20, Max_Features = Auto, Min_Samples_Split = 5, N_estimators = 2000
Bernoulli Naïve Bayes	<pre>param_grid = { 'alpha': [0, 0.01, 0.1, 10, 50, 100], 'binarize': [0.0, 0.5, 1.0, 1.5, 2.0, 2.5, 3.0], 'fit_prior': [True, False], 'class_prior': [np.array([0.5,0.5]), None] } </pre> Fitted 3 folds for each of 168 candidates, totaling 504 fits.	Alpha = 0, Binarize = 0.0, Class_prior = array([0.5,0.5]), Fit_prior = True

Gradient Boosting	<pre>param_grid = { 'n_estimators': [100, 200, 300, 400, 500, 600, 700, 800, 900, 1000] 'max_depth': [1, 2, 5, 10, 20], 'learning_rate': [0.001, 0.01, 0.1, 1.0] } Fitted 3 folds for each of 300 candidates, totaling 900 fits.</pre>	<pre>N_estimators = 100 Max_depth = 20 Learning_rate = 1.0</pre>
-------------------	--	--

Table 1 – Hyperparameters of Machine Learning Algorithms Included in Grid Search

However, the optimal hyperparameters identified in the grid search did not generalize well to the different permutations of our models. The “optimal” hyperparameters did not enhance the accuracies of our models in individual permutations (HD vs. Severe, HD vs. Moderate, HD vs. Mild). The generalization was particularly not good for CD8 models, whereas marginal improvements were found in CD4 models, so we left the hyperparameters as is in the CD8 models, while adjusting the hyperparameters in the CD4 models where improvements were observed. Notably, during holdout testing, the hyperparameters specified in the table above did not improve the performance of our models on the test set of our data in the CD8 data. Therefore, we did not find a compelling reason to adopt the entire set of new hyperparameters specified by results of the grid search, and selectively adopted them as warranted by improvements to model performance.

The parameters that were used in our final reported models are the following.

Algorithm	Final Hyperparameters (CD8 models)	Final Hyperparameters (CD4 models)
SVM	kernel='poly', C=20, degree=5, probability=True	kernel='rbf', C=100, gamma=1, probability=True
KNN	K = 3 weights='uniform' algorithm='auto' leaf_size=30 p=2 metric='minkowski'	K =3, weights='distance', algorithm='auto' leaf_size=10, p=2 metric='minkowski'
Random Forest	n_estimators=100, criterion='gini', max_depth=None, min_samples_split=2, max_features='sqrt', max_leaf_nodes=None, bootstrap=True	n_estimators = 2000, criterion='gini', max_depth = 20, min_samples_split = 5, max_features='sqrt', max_leaf_nodes=None, bootstrap=True
Bernoulli Naïve Bayes	Alpha = 1.0 Binarize = 0.0 Fit_prior = True Class_prior = None	alpha = 0, binarize = 0.0, fit_prior = True class_prior = np.array([0.5,0.5])
Gradient Boosting	n_estimators=100, learning_rate=1.0, max_depth=1	n_estimators=100, learning_rate=1.0, max_depth=20

Table 2: Final Table of Hyperparameters

Page 4 lines 9-10: It might be helpful to explain why different thresholds are used for different datasets.

We thank the reviewer for this comment. The reason for the different thresholds used for the AB and ISB-S datasets is due to the relative sizes of the datasets and the different relative contributions of individual CDR3 sequences to the overall proportions due to sequencing depth; moreover, as the AB dataset is more than an order of magnitude larger than the ISB-S dataset, we chose to use the proportion threshold of 0.0001 for ISB-S samples and 0.00001 for AB samples, so that we analyze an overall comparable number of sequences in each dataset. These different thresholds are necessary given the differences between bulk and single cell TCR sequencing.

Specific minor comments:

Page 7 lines 11-13: This sentence doesn't make sense to me: "Comparison with the top enriched motifs found from the GLIPH2 analysis, including AGQGA%E, S%AAG, SL%AG, SLQGA%YE, S%SGTDT, SL%GTD, SLS%TDT, and S%AGNQP revealed high density of clusters in cluster 6"

Page 11 lines 21-22: This sentence doesn't make sense to me (either semantically or grammatically): "For the ISB-Swedish cohort, patients were first filtered by those were sequenced by 10X Genomics."

Page 13 line 32: Used acronym SVC instead of SVM. I think support vector machines have been mixed up with the name of the sklearn function, support vector classification, which may be confusing to the reader.

We thank the reviewer for these comments. The sentences has been modified in the manuscript for clarity. The acronym has been modified in the manuscript to "SVM."

Reviewer #2:

This article describes an integrative and immunology approach to systematically study more than 4 million TCR sequences from different sources for COVID-19 patients and healthy donors. Overall, this is a valuable effort that helps revealed patterns of the adaptive immune response during SARS-CoV-2 infection. In addition, through the analysis of biological pathways, the authors also suggested that T cell clonal expansion is highly related to some T cell effector functions and TCR signaling. However, some issues or concerns need to be addressed.

Response:

We thank the reviewer for an overall accurate summary and positive comments on the importance and novelty of the project. We address specific comments below:

1. For k-mers analysis, why did the author set 50,000 top variance unique k-mers for downstream analyses e.g., PCA and machine learning pipelines. Will different threshold settings bring different clustering and performance?

For the k-mers analysis, we selected the 50,000 top variance unique k-mers to 1) keep the data dimensions consistent, 2) increase the efficiency of the pipelines, and 3) use data features that are most likely to be the most biologically meaningful. As the -mer increases, the total number of unique kmers in the dataset increases—the likelihood of observing a common 3mer string is much higher than the likelihood of observing a common 6mer string, which means we observe a greater number of unique 6mers than 3mers. However, not all of these 6mers are expected to be meaningful immune signatures. Low-variance kmers, either due to lack of representation (i.e. only in one patient or one T cell clone) or due to conservation between healthy and disease samples are unlikely to be significant in the T cell response to COVID given that the signature is not shared or enriched by disease status. Highly variable kmers, on the other hand, may be reflective of T cell clones that are expanded because of infection, and may also capture correlative information about the clinical severity of symptoms and the T cell signatures of infection.

By reducing the number of kmers that we study in our analysis based on variance, we filter out data that is likely uninformative. It is the same intuition for performing dimensionality reduction prior to machine learning analyses to first identify the most meaningful features of the data. We also keep the data dimensions consistent by using this threshold. Changing the inclusion threshold is unlikely to show different clustering or performance. Including more low-variance kmers by applying a more lenient threshold is unlikely to capture more biologically meaningful signals; reducing the kmers by applying a more stringent threshold will further tailor the analyses to the highest variance kmers, which are features that the machine learning analysis is likely detecting.

2. In addition to K-mers, did the author consider using other methods, such as deep learning, multiple sequence alignments and other methods for clustering and other downstream analysis?

We thank the reviewer for this comment. Yes, we have tried methods such as multiple sequence alignments and deep learning before trying our kmer-based methods. One approach that we attempted

was: 1) multiple sequence alignment on all TCR CDR3 sequences, 2) representation of each patient’s TCR repertoire as a matrix of one-hot encoded, MSA-aligned vectors of CDR3 sequences, like an image, and 3) training a convolutional neural network (CNN) to classify each patient’s TCR repertoire as HD or COVID. This approach was similar to the CNN-based algorithm described by Beshnova et al. (2020) to detect cancer-associated T cell receptors, where TCR sequences were similarly represented as data matrices the way images are represented prior to training a CNN⁷. While this approach initially appeared promising for our research question, we did not observe high classification accuracy, so we opted not to present these findings in our manuscript. We believe that this approach was unsuccessful because the CNN was overfitting on the training data. Representing the TCR data as one-hot encoded matrices following MSA alignment likely introduced a lot of “zeros” as MSA allowed for “gapped” alignments, and all such “gaps” were treated as zeros in the one-hot encodings — consequently, the data representation may have been too sparse for the algorithm to learn meaningful features. Below are the original plots of the ROC curve, training vs. validation losses, and training vs. validation accuracies.

Training and Validation loss: Healthy Donor VS Severe, MSA encoded TRA_CDR3 + TRB_CDR3

Training and Validation accuracy: Healthy Donor VS Severe, MSA encoded TRA_CDR3 + TRB_CDR3

Performance of the Convolutional Neural Network Classifier (HD vs. COVID) trained on Multiple-Sequence-Aligned TCR CDR3 Sequences

⁷ Beshnova, Daria, et al. “De Novo Prediction of Cancer-Associated T Cell Receptors for Noninvasive Cancer Detection.” *Science Translational Medicine*, vol. 12, no. 557, 2020, <https://doi.org/10.1126/scitranslmed.aaz3738>.

3. In the machine learning part, because the number of healthy donor samples is relatively small, even though the author uses the unsampled method, is there any possibility of overfitting? Because I have observed that the performance of many classifiers is 1. I suggest that the model performance can be re-evaluated on the independent test set collected separately.

We thank the reviewer for this comment. Upon re-examination of our code, we discovered that during the process of up-sampling the data and subsequent test-train set partitioning, we generated train and test sets that contained the same data due to the up-sampling procedure (which was done to balance the number of COVID and HD samples in the testing and training sets). The reviewer is correct that this resulted in overfitting of our models. In our revision, we have corrected for this methodological error by partitioning our data into test and train sets prior to balancing the datasets. This way, we have ensured that no duplicate data is present in both test and train sets. In our new figures, our algorithms are being trained on data that is distinct from the data that the algorithms' performance is evaluated on.

We also agree with the reviewer that it would be valuable to train our algorithms on one dataset, and then test performance on a completely independent dataset, as this external validation set will prove the generalizability of our machine learning models to multiple TCR datasets. We included above the results of several new machine learning models that we trained on the ISB-S dataset that we subsequently tested on the PLA and WHH datasets as external validation sets.

4. Regarding the predictor, the author did not release an independent program or online predictor, so it is not available for others to test or use.

We thank the reviewer for this comment. We have included in our resubmission the refactored code with our final trained models that can be evaluated by user-defined input files.

5. Another key question is whether there is any clinical application of the findings in the author's article. What is the point of finding these important TCR patterns? Such as the development of personalized vaccines, observation of potential COVID-19 patients.

We thank the reviewer for this comment. In the context of COVID-19, our findings elucidate distinct clonal expansion patterns and TCR repertoire features in patients experiencing symptoms of varying clinical severity, and potential TCR sequence motifs of interest. With further validation studies, these motifs present in clonally expanded T cells may serve as prognostic and diagnostic biomarkers in COVID infection. Sequence motif studies can be of great clinical significance in broad contexts. Immune cell receptor motif-based investigations are increasingly become high utility, where systematic investigations of the specific motifs in the CDR3 region have identified T cell clones associated with specific immune

functions like gluten hypersensitivity in celiac disease⁸, hyperinflammation in Ankylosing spondylitis⁹, and reactivity to cancer neoepitopes¹⁰. As longitudinal studies further uncover physiological and immunological effects of long COVID, where patients experience long-term adverse effects from past COVID infection, it is of potential interest to compare the TCR repertoire profiles of these patients to the TCR repertoire features that have been identified in our manuscript.

Broadly, immune repertoire analysis has become a fundamental tool to understand the biology of immune-mediated diseases as well as immune responses to therapies¹¹. T cell receptor motifs have been used as prognostic and diagnostic biomarkers. For example, TCR repertoire studies have shown that improved TCR diversity is linked to better prognostic outcomes for cancer patients who receive immunotherapies^{12 13}. TCR sequencing was used to predict patient prognosis in melanoma patients who received anti-PD1 checkpoint blockade therapy: increased diversity of T cell receptors in patients was linked to better outcomes in terms of tumor progression and progression-free survival¹⁴. This analysis is similar to the analysis we have conducted in our manuscript, where we show the effect of COVID-19 infection in the clonal diversity of T cells. Moreover, TCR repertoire sequencing revealed an increase in hyperexpanded TCR clonal frequencies following the administration of a neoantigen vaccine, elucidating the role of T cell dynamics in tumor immunology¹⁵. In light of these findings, similar TCR studies in the context of COVID-19, with a particular emphasis on convalescent patients upon treatment with investigational COVID therapies such as Nirmatrelvir or Bebtelovimab are of potential interest, to study the T cell dynamics in investigative COVID treatments.

6. The author constructed 3-mer, 4-mer, 5-mer, and 6-mer frequency matrix representations of ISB-S CD4 and CD8 datasets and performed PCA analysis to see whether samples cluster by disease severity. I observed that the majority of samples clustered together, although a number of mild and moderate samples were separated from the main cluster. So, is it possible that this part of the separated samples is

⁸ Dahal-Koirala, S., Risnes, L., Neumann, R., Christophersen, A., Lundin, K., Sandve, G., Qiao, S. and Sollid, L., 2021. Comprehensive Analysis of CDR3 Sequences in Gluten-Specific T-Cell Receptors Reveals a Dominant R-Motif and Several New Minor Motifs. *Frontiers in Immunology*, 12.

⁹ Zheng, M., Zhang, X., Zhou, Y., Tang, J., Han, Q., Zhang, Y., Ni, Q., Chen, G., Jia, Q., Yu, H., Liu, S., Robins, E., Jiang, N., Wan, Y., Li, Q., Chen, Z. and Zhu, P., 2019. TCR repertoire and CDR3 motif analyses depict the role of $\alpha\beta$ T cells in Ankylosing spondylitis. *EBioMedicine*, 47, pp.414-426.

¹⁰ Bravi, B., Balachandran, V., Greenbaum, B., Walczak, A., Mora, T., Monasson, R. and Cocco, S., 2021. Probing T-cell response by sequence-based probabilistic modeling. *PLOS Computational Biology*, 17(9), p.e1009297.

¹¹ De Simone, M., Rossetti, G. and Pagani, M., 2018. Single Cell T Cell Receptor Sequencing: Techniques and Future Challenges. *Frontiers in Immunology*, 9.

¹² Hogan, Sabrina A., et al. "Peripheral Blood TCR Repertoire Profiling May Facilitate Patient Stratification for Immunotherapy against Melanoma." *Cancer Immunology Research*, vol. 7, no. 1, 2018, pp. 77–85., <https://doi.org/10.1158/2326-6066.cir-18-0136>.

¹³ Hosoi, A., Takeda, K., Nagaoka, K. et al. Increased diversity with reduced "diversity evenness" of tumor infiltrating T-cells for the successful cancer immunotherapy. *Sci Rep* 8, 1058 (2018). <https://doi.org/10.1038/s41598-018-19548-y>

¹⁴ Poran, Asaf, et al. "Combined TCR Repertoire Profiles and Blood Cell Phenotypes Predict Melanoma Patient Response to Personalized Neoantigen Therapy plus Anti-PD-1." *Cell Reports Medicine*, vol. 1, no. 8, 2020, p. 100141., <https://doi.org/10.1016/j.xcrm.2020.100141>.

¹⁵ Poran, Asaf, et al. "Combined TCR Repertoire Profiles and Blood Cell Phenotypes Predict Melanoma Patient Response to Personalized Neoantigen Therapy plus Anti-PD-1." *Cell Reports Medicine*, vol. 1, no. 8, 2020, p. 100141., <https://doi.org/10.1016/j.xcrm.2020.100141>.

caused by sequencing technology or batch effects? I think the conclusion the author got here is not supported by statistical significance?

We thank the reviewer for this comment. It is not possible to attribute the separation of these mild and moderate samples to batch effects or sequencing technology, as all the data being shown in the PCA plots were generated from the same “batch.” To clarify what is being shown, dimensionality reduction in this plot was performed only the ISB-S dataset, all of whose samples were sequenced under uniform protocol and processed identically. While such batch effects may be a relevant concern in a dimensionality reduction plot that integrates samples from differing wet lab protocols and sequencing technologies, we do not believe that this is a relevant concern here.

Reviewer #3: This paper is a meta-analysis of TCR data from COVID-19 and healthy controls. The authors bring together data from a number of previous studies, and provide the single biggest set of TCR data from COVID-19 patients. The paper makes a number of claims about the COVID-19 repertoires. As in all such metadata, especially where the data were produced by completely different methodologies, it is absolutely crucial to provide full clarity about the composition of the different data sets. One key element is to show how many of each cohort were mild, moderate or severe, or HD. I may have missed this somewhere in the supplementaries, but it is essential to make this data obvious in fig 1. I suspect the cohorts are very unbalanced.

Response:

We thank the reviewer for an overall accurate summary of this project. We agree that it is important to clarify the nature of the datasets in our meta-analysis. The four datasets that are used in our manuscript are delineated in Figure 1A: 1) Adaptive Biotechnologies, 2) ISB-Swedish COVID-19 Biobanking Unit, 3) PLA General Hospital and 4) Wuhan Hankou Hospital. Regarding the breakdown of how many of each cohort were mild, moderate, severe, or HD, we provide a table of counts below, labeled “Table 2,” which has also been included in Figure 1 in the resubmission. The reviewer is correct that the cohorts are not balanced. However, despite the imbalanced data, the large sample size of the Adaptive Biotech and ISB-S datasets enable statistically meaningful analyses of all categories of COVID severity and a meaningful comparison to the healthy donors.

Dataset	HD Count	Mild Count	Moderate Count	Severe Count	Convalescent Count
AB	88	1475			0
ISB-S	16	108	93	49	0
PLA	5	0	7	4	6
WHH	5	0	5	5	0

Table 3 – Tabulated Summary of Number of Patients in Each Category per Dataset

In fig 1, all the estimators are strongly influenced by sample size. The authors should show their conclusions are not influenced by sample size. I could not understand what panel E represented. In F, was this just a random sample of 32 AB repertoires? Why 32? Was this repeated multiple times?

We thank the reviewer for this comment. We acknowledge that in Figure 1B, which shows comparisons of clonal diversity between HD and COVID TCR repertoires, the p-values of the differences are inevitably influenced by sample size. For this reason, the figure also presents a more holistic visualization of the boxplot with individual data points, so that the basis of comparison is not solely the p-value. As AB, ISB-S CD4, ISB-S CD8 datasets have much greater sample size than the PLAGH and WHH datasets, even with similar effect sizes, the larger datasets are expected to generate a much smaller p-value. It is not surprising that even with large effect size, the p-values are not as statistically significant in the smaller datasets, PLAGH and WHH. Nonetheless, we do not believe the different sample sizes diminish our general claim that on average, COVID patients have TCR repertoires with lesser clonal diversity than healthy donors. Even without considering the p-value, visualization of the effect size appears significant in all datasets except perhaps in AB, where we observe much higher variance in the clonal diversity of the

COVID group. Perhaps this variance is intrinsically interesting, even if it makes the effect size less clear — while healthy donors have similar degrees of clonal diversity, COVID patients vary widely in the AB dataset.

We believe that the greater number of outliers in just the COVID group across all datasets in Figures 1B and 1C, and the wider spread in clonal diversity in the COVID group compared to HD group, are interesting findings, even if the mean comparisons are not as convincing to the reader. They reflect heterogeneity in T cell repertoire changes upon COVID infection, the directionality of which is generally a decrease in diversity. We have also included multiple metrics of clonal diversity in our manuscript to support the claims being made in Figure 1B. Figure 1C uses an alternate metric, the Gini-Simpson index, to support this claim. The boxplot visualizations in 1C, which include the distribution of individual data points, support our observation even in the datasets where the p-value is not statistically significant — in the WHH and PLAGH datasets, where the p-values are not statistically significant at the typical threshold of $\alpha = 0.05$, visually, we still see a clear difference in the Chao diversity metric between healthy donors and COVID patients. While sample size inevitably affects the p-value, we believe that the general claim of this figure regarding clonal diversity still stands.

We apologize that Figure 1E was not clear to the reviewer. Figure 1E is showing that the CDR3 sequences enriched in the COVID patients had significant overlap (among mild, moderate and severe patients), while there was almost no overlap in CDR3 enrichment between healthy donors and covid-19 patients. The figure legends have been updated for clarity. Regarding Figure 1F, yes, we are showing a random sample of 32 AB repertoires each from HD and COVID-19. We deemed this sample to be representative of all of the repertoires because repeated sampling generated very similar plots. We chose to visualize a sample of the healthy donor and COVID repertoires because it is not feasible to show properties of the entire repertoire in such a visualization due to the sheer size of the dataset. Also, because the dataset is imbalanced, this random sampling allows us to show representative numbers of healthy donor and COVID repertoires for visualization purposes. Overall, we aim to show in Figure 1F that the overrepresentation of specific clonotypes in COVID-19 patients is consistent with the observation of reduced clonal diversity in Figures 1B and 1C.

In Fig 2, the PCAs are not convincing to me. There are a few outlying mild and moderate, but otherwise they look totally overlapping.. In the heatmaps, are these averages of multiple subsamples ? Is there any obvious separation between condition if the samples are allowed to cluster across the different individuals ? I did not really understand the import of the panels G and H. Could the authors explain this a bit more clearly ?

We thank the reviewer for this comment. In the PCA plots in Figure 2, we agree that it is difficult to discern discrete clusters by disease severity. However, we hope to show in this figure that there are a greater number of outliers in mild and moderate disease compared to HD or severe, even though we cannot discern clear clusters by severity.

In retrospect, it is not surprising that the data cluster together because there are general rules that define many CDR3 amino acid sequences that are common to all individuals. For example, “CASS” are four amino acids that are commonly found at the beginning of many CDR3 sequences regardless of the

individual, so this motif will be highly represented in the repertoires of healthy donors and COVID patients alike. Similarly, numerous CDR3 sequences share an “end” motif, which also contributes to the homogeneity of motif signal. In other words, many people will share kmer motifs that are part of the “governing motifs” of CDR3 sequences like the “CASS” start motif. Given this information, upon dimensionality reduction based on kmer data, this homogenous signal is likely to dominate the heterogeneous CDR3 kmers that differentiate individuals’ repertoires.

In light of this information, perhaps the outliers in the PCA that fail to fall into the large central cluster are more interesting — the PCA is detecting from the kmers in these patients’ repertoires that these outliers harbor high-variance data features. Rather than looking at the “bulk signal,” which is the large central cluster, in the outliers that do not fall into this cluster, we may be observing the rare variations in CDR3s that have “signal” that is differentiated by unique enrichment of these sequences. Notably, these data points come from mild and moderate patients, rather than severe disease patients, which may suggest that as part of mounting an effective adaptive immune response to COVID infection, the T cell repertoires may be undergoing changes that selectively enrich certain clones that harbor specific TCR motifs, which are being captured in the PCA plot.

We acknowledge that we cannot expect to observe clear, discrete clusters by disease severity in the PCA due to the intrinsic properties and similarities of CDR3 amino acid sequences. However, perhaps the signal of outliers in the PCA comprising only mild and moderate repertoires is potentially interesting.

In Fig 3, what is the proportion of cells between different disease subsets? Can the authors say a bit more about the subsets? Which are naïve? Is there a proliferating subset? The authors say there is a relationship between the cluster pattern and disease severity – but it looks more like a radically different distribution between healthy and COVID-19 repertoires. This seems a striking finding, and not necessarily what other people have observed? This figure deserves a much more detailed discussion and analysis.

We thank the reviewer for this comment. Here is a summary table of the cells shown in Figure 3.

Disease Condition	Sum of Cell Count	Proportion of Cells
HD	14096	10.33%
Mild	50625	37.11%
Moderate	47067	34.50%
Severe	24641	18.06%
Grand Total	136429	100%

Table 4: Cell Counts in scRNAseq UMAP Plots in Figure 3

Figure S5 shows some gene markers that may be informative for the reviewer. The proliferating subset is shown in the UMAP plot in Figure S5B, in the supplemental figures, where we indicate the clonally expanded cluster in red. Putting Figures 3 and S5 side by side, we can juxtapose the clonally expanded T cells containing the COVID-enriched motifs shown in Figure 3C with the differential gene expression shown in Figure S5D. GNLY encodes the cytotoxic granules of T cells, which are released upon antigen stimulation, and it is highly expressed in the cells of cluster 6. This shows that the cells of cluster 6 are

activated, proliferating cytotoxic T cells, which is consistent with what we expect from the clonally expanded cells containing COVID-enriched GLIPH2 motifs that are shown in Figure 3C and Figure 3B.

Juxtaposition of Figure 3 and Figure S5: Proliferating Subset

The naïve subset is shown in the UMAP plot of the naïve phenotype-related markers TCF7 and LEF1, where the naïve subset corresponds to the cells marked in yellow / orange in the plots below (high expression of TCF7 and LEF1 indicates naïve phenotype). Comparing all of these plots below at first glance, we observe that there is an abundance of naïve subset in the “healthy donor” T cell section of the UMAP, whereas we do not observe many naïve T cells that are in cluster 6, which is the clonally expanded cluster with COVID-enriched motifs.

Juxtaposition of Figure 3 and Figure S5: Naïve Subset

Regarding the reviewer’s comment that Figure 3 appears to point more to a radically different distribution between HD vs. COVID patients as opposed to cluster patterns by disease severity, we agree with the reviewer that we observe a very salient contrast between healthy donors and COVID patients (especially in Figure 3B), but also we see that while moderate and mild cells are enriched in the center of cluster 6,

the severe cells are enriched in the rightmost corner of cluster 6, suggesting slight differences in the clustering across disease severity states. Nonetheless, we agree with the reviewer that the cells cluster much more saliently between HD vs. COVID, and the differences across COVID severities is not quite as obvious.

Figure 3B, for the reviewer’s convenience.

Fig 4 is lacking a lot of detail which is important for interpretation. How balanced are the repertoires ? What is meant by upsampling the controls ? How were the parameters decided – was this done on a independent data set (i.e. before the cross-validation) ? Did the authors try fitting models on one data set, and then testing on one of the other data sets ?

We thank the reviewer for this comment. In our revision, we have clarified the number of samples in each category in each of the datasets in Figure 1 of the manuscript, and in Table 2 above in this response letter. As the reviewer pointed out in an earlier comment, our datasets are not balanced with respect to the number of healthy donors vs. mild COVID vs. moderate COVID vs. severe COVID. Notably, there are fewer healthy donor TCR repertoires in the ISB-S dataset (which is the dataset that all of our kmer machine learning models were constructed on) compared to the number of TCR repertoires in all disease classes.

This imbalance was the reason for upsampling the controls, by which we mean duplicating the number of Healthy Donor sequences such that the number of HD repertoires in the training set matches the number of COVID repertoires in the training set. Most machine learning models do not perform well when there is a significant imbalance in the data. In imbalanced classification problems, algorithms often generate biased predictions that predict all test set data as the over-represented category, or the “majority class.” There are several ways to address this issue. One approach is to take a subsample of the overrepresented category of the dataset such that it matches the amount of data in the underrepresented category. Another approach, which is the approach we take, is to resample the data such that data in the underrepresented category is randomly duplicated to match the amount of data in the overrepresented category. A key advantage of the oversampling approach is that we do not discard any data during the training of our machine learning algorithms. Having a high sample size is a critical issue in attaining good machine learning performance. Due to sample size concerns, we chose our approach of balancing our training data via oversampling the minority class, as opposed to under-sampling the majority class.

Regarding the hyperparameters of the model, we initially opted for default parameters provided by the scikit-learn library. However, as explained above, we subsequently tried a more rigorous approach in selecting the hyperparameters by performing a grid search of several hundreds of hyperparameter combinations. This grid search was performed on the combined ISB-S CD4 and ISB-S CD8 datasets for classification of HD vs all COVID repertoires (6mer data). Nevertheless, as explained previously, the “optimal” hyperparameters identified through this grid search did not substantially affect or improve the models’ performance during holdout testing of individual permutations (HD vs. Mild COVID, HD vs. Moderate COVID, HD vs. Severe COVID). Therefore, the hyperparameters of our models are as written in our methods section, as there was no clear justification for opting for a different set of parameters than the default.

We have explored alternative methods to try fitting models on one data set, and then testing on one of the other data set. To evaluate whether machine learning algorithms trained on T cell repertoire data are generalizable, we have tested several data curation methods and algorithms that allow us to train our models on the ISB-S dataset and test performance on the PLA and WHH datasets.

While our kmer models were not generalizable to other datasets, we successfully implemented an alternative data curation approach using motif-based clustering, which enabled the generalizability of our machine learning models to external validation sets. While the different datasets were unlikely to include identical kmers, we hypothesized that CDR3 motifs would still be found in common across datasets, so we opted to curate our data as a motif-based counts matrix using GLIPH2 motifs instead of kmers. With this new data curation approach, we were able to train our machine learning models on the ISB-S dataset and assess performance on the PLA and WHH datasets as an external hold-out testing validation. We trained the models on two permutations (HD vs. Moderate and HD vs. Severe) on the ISB-S dataset and tested the models on the equivalent subsets of the PLA (Zhang) and WHH (Wen) datasets. As the PLA and WHH datasets lack any mild COVID patients, we did not try this permutation. We also built soft-voter ensemble classifiers of the five classifiers (Random Forest, Support Vector Machine, Gradient Boosting, KNN and Bernoulli Naïve Bayes), which generated highly accurate predictions in the validation sets. The results are reported below.

GLIPH2 Models: HD vs. Severe, Train = Su, Test = Wen

GLIPH2 Models: HD vs. Severe, Train = Su, Test = Zhang

GLIPH2 Models: HD vs. Moderate, Train = Su, Test = Wen

GLIPH2 Models: HD vs. Moderate, Train = Su, Test = Zhang

**ROC Curves of GLIPH2-based models, Five Supervised ML Classifiers:
External Validation Set Holdout Testing Results –
Training on ISB-S, Testing on PLA (Zhang) and WHH (Wen)**

Soft Voter Ensemble: Test = Wen Severe (AUC=0.72)

Soft Voter Ensemble: Test = Zhang Severe (AUC=1.00)

Soft Voter Ensemble: Test = Wen Moderate (AUC=0.84)

Soft Voter Ensemble: Test = Zhang Moderate (AUC=0.71)

**ROC Curves of GLIPH2-based models, Soft Voter Ensemble Classifiers:
External Validation Set Holdout Testing Results –
Training on ISB-S, Testing on PLA(Zhang) and WHH (Wen)**

Overall, the attempt to provide a meta-analysis of diverse TCRrep data sets is interesting and potentially valuable. But in light of the known challenges of meta-analyses, the paper is a bit skimpy in providing detail which could be used to assess the validity of the comparisons. Also, at the end of the day, each analysis has largely been carried out on an independent data set, which rather reduces the value of carrying out a meta-analysis in the first place. Other than the overall reduction in diversity shown in fig 1, I am not convinced any of the other findings are generalizable across data sets, and so its hard to assess the significance of the findings.

We thank the reviewer for this comment. We acknowledge that our study has limitations due to the design of our analyses, which have more extensively focused on the ISB-S dataset compared to the other three datasets included in our manuscript. The primary reasons for devoting more extensive analyses on the ISB-S dataset are 1) its large sample size and 2) availability of both scTCR-seq and scRNA-seq data, which allowed for our study to relate TCR motif-based insights to gene expression data and thereby integrate several phenotypic information together. This depth of analyses was not possible for the other datasets — nonetheless, as this analytical approach is novel and not previously described in the literature, we still believe that our findings positively contribute to the body of COVID-19 research. The AB dataset

offered other advantages, such as even larger sample size, which enabled high-powered, broader insights about changes to T cell clonal dynamics — although it was not possible to conduct single-cell transcriptomics analyses and identify individual T cell populations of interest as it is a bulk sequencing dataset. We acknowledge that fewer of our analyses have focused on the PLA and WHH datasets, although the analyses in Figure 1 still illuminate several consensus results of TCR repertoire features in COVID that are shared by PLA and WHH datasets, which have served to add validity to our findings. The less extensive focus on the PLA and WHH datasets was an unfortunate limitation of the small sample size of these datasets and the unique challenges of performing integrative analyses in scTCR-seq data.

A major reason for our largely separate analyses pipelines conducted on individual datasets, rather than attempting a pooled, integrative analysis, is that the datasets in our manuscript are not highly comparable at baseline, and integrative analyses would require additional challenging considerations of batch effect correction. As the reviewer points out, meta-analyses may apply analyses pipelines uniformly on an integrative dataset that combines data from multiple sources. Notably, this approach may be feasible if all datasets are single cell immune repertoire sequencing data and are also meaningfully comparable in terms of the cell types included and laboratory methods employed. Unfortunately, this approach was not possible for our manuscript, because the datasets in this manuscript includes a bulk-TCR sequencing dataset (Adaptive Biotechnologies), which cannot be readily integrated with the single-cell datasets (ISB-S, PLA-H, WHH).

It is, in theory, possible to perform integrative and pooled analyses for the three single-cell datasets if we were primarily interested in gene expression via RNA-seq. However, our analysis is focused on TCR-seq data from the multiple datasets, which has unique challenges compared to RNA-seq data when considering the possibility of batch effect correction. While scRNA-seq quantifies expression levels of genes, where intuitively, some baseline signal of the “batch” from each dataset can be subtracted from the background to separate “signal” from “noise,” in scTCR-seq, the data structure is comprised of sequence information rather than quantifiable expression data. We therefore focused our analyses on gaining insights into changes in repertoire diversity, specificity, and clonal composition within particular datasets.

Additional considerations that reinforced our decision to pursue our analytical approach were that the success of computational approaches for batch effect correction is the most salient among datasets with high baseline similarity. The datasets in our manuscript lack such a high degree of baseline similarity. Crucially, the ISB-S dataset used droplet-based sequencing for the TCR sequences, whereas the PLA and Wuhan Hankou dataset used full-length sequencing, which reduces the baseline comparability of the datasets in the first place. Also, the three single-cell datasets are not of equal size. The ISB-S (Su) dataset vastly outnumbers both the WHH (Wen) and PLA Hospital (Zhang) datasets in sample size. If a pooled analysis approach is undertaken, we anticipate that the biological signal from the ISB-S dataset will be disproportionately amplified in the results, compared to the biological signal from the PLA and WHH datasets. Even if we were to obtain scRNA-seq data of the WHH and PLA datasets to generate a pooled scRNA-seq UMAP of WHH, PLA and ISB-S T cells, we would generate a scRNA-seq plot that is so highly dominated with cells from the ISB-S dataset that it would be difficult to obtain any signal from WHH or PLA datasets. This can also be intuited from the table of cell counts provided in Table 4 of this response letter.

As such, we believe that an analytical pipeline that separately analyzes the individual datasets and reports the results from each dataset as well as the consensus results, which is the approach in our manuscript, is more methodologically sound than attempting a pooled approach. Regarding the generalizability of the machine learning models, we investigated using one dataset for training machine learning models and testing their performance on other datasets as described in the previous sections.

Reviewers' comments:

Reviewer #1 (Remarks to the Author):

Given the significant amount of helpful additional explanation in the 26 page rebuttal letter, I am genuinely surprised how little of this explanation has found its way into the text, which has not changed much. For the majority of readers, I think it will remain exceptionally hard to understand what has been done here. I agree with the comments of the other reviewers in their round 1 reviews. Unfortunately, I am not clear that many of their concerns have been addressed in the manuscript either, although they have been addressed at length in the rebuttal letter.

Reviewer #2 (Remarks to the Author):

The author addressed my concerns very well. I have no more comments.

Reviewer #3 (Remarks to the Author):

The underlying problems associated with this analysis remain. The datasets are highly unbalanced. The size of the repertoires will influence the diversity measurements - this has nothing to do with p-values, but arises from the distributions of TCR frequencies which are not linearly scalable. However the authors have added significant additional information and clarification so that the readers may be aware of these limitations, and take them into account in evaluating the results.

Response letter for revision of Park et al., “Machine learning identifies T cell receptor repertoire signatures associated with COVID-19 severity”

Summary of this revision

- Incorporated suggested changes into the new manuscript with page and line numbers to indicate their locations.
- Acknowledged important limitations of the study in the Discussion section (pages 11-13), including the limited generalizability of kmer-based machine learning models, unbalanced datasets, and the limitations as a meta-analysis due to separate analyses pipelines on each dataset as opposed to an integrative, pooled approach.
- Ensured no sample overlap between training and validation datasets (pertinent to Figure 5) by fixing an error in upsampling process
- Acknowledged the sample size-related limitations in our findings regarding COVID-related clonal expansion/diversity reduction in Figure 1B-C.
- Elaborated upon the differences in wet-lab sequencing methodologies in each of our four datasets to explain why a typical meta-analysis approach of pooled, integrative analysis was not feasible in our study.
- Reworded a number of sentences and defined a few terminologies to avoid ambiguity and improve readability.
- Described the sample size in each dataset/group as well as the optimization process of parameters of machine learning models.

Details to follow below.

Point-by-point response to the reviewers (Park et al., *Communications Biology* manuscript)

Reviewer 1 comments

Given the significant amount of helpful additional explanation in the 26-page rebuttal letter, I am genuinely surprised how little of this explanation has found its way into the text, which has not changed much. For the majority of readers, I think it will remain exceptionally hard to understand what has been done here. I agree with the comments of the other reviewers in their round 1 reviews. Unfortunately, I am not clear that many of their concerns have been addressed in the manuscript either, although they have been addressed at length in the rebuttal letter.

We thank the reviewer for this feedback. In this revision, we have focused on incorporating the responses in the previous rebuttal letter into the body of the manuscript. Here, we also include the page numbers of the incorporated changes, and in some cases, the direct quotes, to highlight the changes in the manuscript. We hope that these changes enhance the clarity of our analyses to the readers.

Reviewer 1 comments from last revision: Key concerns are (1) a lack of consideration of bias due to different laboratory sequencing methodologies for the different cohorts, which have been drawn together, and (2) the extent to which the same samples were used in both training and testing sets, leading to an overestimate of the performance of the machine learning methods.

Last revision, we acknowledged that the reviewer makes a great point regarding the differences in laboratory sequencing methodologies among the four datasets. We further explained in our response letter that these differences have precluded the possibility for integrative, pooled analysis in our manuscript, which is the reason why we opted to conduct a separate analysis of each dataset independently and report consensus findings. Our analysis approach has thus circumvented the need to consider these sequencing biases and batch effects. In this revision, we make this important methodological distinction clearer in the main body of the manuscript in both the methods (page 15) and discussion sections (page 13).

- This paragraph was included on Page 15 (Methods): “The Adaptive Biotechnologies dataset comprised bulk TCR-seq data, while the ISB-S, PLA, and WHH datasets comprised single-cell TCR-seq data, with variations in sequencing modalities, patient populations, and sample sizes. Due to the differences in wet lab protocols and the potential presence of batch effects in each of these four datasets, all downstream analyses were performed separately on each individual dataset, and the result from each dataset as well as the consensus findings are reported.
- This paragraph was included on Page 13 (Discussion): “Finally, while a pooled integrative analysis approach is preferable in a meta-analysis to show the generalizability of findings across datasets, this approach was impractical for our study due to the lack of baseline comparability including variations in sequencing modalities, patient populations, and sample sizes. Due to differences in data collection protocols among the four datasets, a pooled analysis approach would have likely introduced significant batch effects that obscure true biological signal. Instead, separate

bioinformatics analyses pipelines were conducted on individual datasets, which allowed all analyses to remain free of batch effects.”

Moreover, in the last revision, we acknowledged and addressed the issue that there was overfitting in our first submission’s machine learning models because of the overlap between training and testing sets. The overlap between training and testing sets was due to an error occurred during up-sampling procedure, which was done to balance the number of COVID and HD samples in the testing and training sets. In our last revision, we have corrected for this mistake in our code and submitted revised figures (Figure 5) that confirm that this is no longer affecting our analyses. We have corrected this methodological error by partitioning our data into test and train sets prior to balancing the datasets. This way, we have ensured that no duplicate data is present in both test and train sets. In our new figures, our algorithms are being trained on data that is distinct from the data on which the algorithm performance is evaluated.

Reviewer 1 comments from last revision: Page 3, lines 29-32: Introduction - I am unsure of the reason for discussing a largely irrelevant, clonal process (lymphoblastic leukaemia), while not properly reviewing the literature in terms of single cell or bulk TCR repertoire sequencing work that has contributed to our understanding of SARS-CoV-2 responses.

In the revised manuscript, we have removed the discussion about clonal process and reviewed the literature related to single cell or bulk TCR repertoire sequencing that has contributed to our understanding of SARS-CoV-2 responses (pages 3-4).

Reviewer 1 comments from last revision: Page 4, lines 19-24: Results – This is a key methodological concern: A series of datasets appear to be “lumped together” without any consideration of potentially differing laboratory methods, whether CD4/ CD8 or all T-cells were sequenced or the amount of starting material (i.e., what numbers of lymphocytes the nucleic acid sample used in sequencing equated to). No consideration appears to be given to potential bias introduced by any of this. The manuscript is very much written from a bioinformatician’s standpoint, without critically considering the “wet lab” methods. If I have misunderstood this, I apologise, but it is not at all clear from the manuscript what has been compared with what at various points in the analyses.

Page 9, lines 15-17 again demonstrate the mindset on page 4, considering only bioinformatic analysis: “By uniformly processing immune sequencing data from multiple cohorts with TCR-seq data, we found that antigen exposure during the course of COVID-19 significantly decreased the diversity of repertoires and reshaped clonal representation.” Uniform bioinformatic processing cannot negate the effects of using potentially mismatched cohorts produced using different laboratory methodology with different biases (please see my comment above). I have not had time to go through the various cohorts, but the authors need to reassure the reader that they are comparing like with like, rather than “apples with oranges”, so to speak.

In the revised manuscript, we have highlighted these considerations regarding the differences in laboratory methods that were used to sequence the four T cell datasets included in our study (pages 12-

13) and further discussed how these differences prompted us to choose a separate and independent analysis of each dataset, while using consistent bioinformatics tools.

Reviewer 1 comments from last revision: Page 4, line 5: “fewer differences were found for the ISB-S CD4 and CD8 datasets when comparing samples from different disease severities to those from healthy donors”. Following from the two concerns above, did the healthy donors’ samples contain sequences from both CD4 and CD8 T-cells? This will dilute any signal obtained.

In the revised manuscript, we have clarified that the healthy donors’ samples contained sequences from both CD4 and CD8 T cells. Also, to avoid comparing “apples with oranges”, we have clarified that the ISB-S dataset was partitioned into the ISB-S CD4 dataset and ISB-S CD8 dataset, in the revised manuscript. This sentence was included on Page 5: “The analyses for the ISB-S dataset were further stratified by cell type (CD4 vs. CD8), separated into ISB-S CD4 and ISB-S CD8 datasets, both of which contain TCR sequences of healthy donors and COVID-19 patients.”

Reviewer 1 comments from last revision: Page 4, lines 7-8: “By comparison, the top CDR3 sequences were different across conditions for both the AB and ISB-S datasets.” Again, is this due to differences between laboratory methodology or a genuine difference between patients? This does not appear to be considered.

We thank the reviewer for this comment. Since samples were only compared within a given dataset, these differences are due to patient differences rather than due to methodology. In the revised manuscript, we reworded this sentence to avoid ambiguity or misunderstanding. This sentence was included on Page 6: “By comparison, the top CDR3 sequences of healthy donor, mild COVID, moderate COVID, and severe COVID TCR repertoires were different within the AB dataset, as well as within the ISB-S datasets (Figures 1D, S3A-B).”

Reviewer 1 comments from last revision: Page 13, lines 28-29: It is very important that the same upsampled data is not present in both the training and testing splits, because in that case the authors would be testing on the training data and artificially increasing the cross-validation performance. The authors should provide reassurance that this isn’t happening. However, from inspection of the code, it appears that they are testing on the training data. This is not an appropriate way to assess a machine learning method’s performance.

The revised submission includes substantially modified code as well as the updated ROC curves in Figure 5 and Figure S7 that demonstrate the results after this issue has been resolved. We corrected the upsampling issue by partitioning our data into test and train sets prior to balancing the datasets, thereby ensuring that no duplicate data is present in both test and train sets. This led to a decrease in AUROC values, but the machine learning based approaches still maintained high performances depending on the dataset and pipeline permutation.

Reviewer 1 comments from last revision: Page 9 lines 2-3: It would be helpful for clarity if results were specifically attributed to the testing data partitions here, if that is the case.

Results were specifically attributed to the specific classification task: for example, one model was trained on 6mers to classify moderate COVID from healthy donors from the ISB-S CD8 TCR sequences. Another model was trained on 3mers to classify mild COVID from healthy donors from the ISB-S CD4 TCR sequences. This sentence was included on page 10: "A total of 12 models were trained, with the permutations varying in 1) classifying different levels of COVID severity (HD vs Mild, HD vs Moderate, HD vs Severe), 2) CD4 vs CD8 T cell receptors, and 3) 3mer vs 6mer representation of the TCR data.

Reviewer 1 comments from last revision: Page 9 lines 6-9: It appears that separate models for CD4 and CD8 analysis are used (which is helpful, in light of the above comments), and therefore the authors cannot use the Adaptive dataset for hold-out testing. However, it would be good practice if they acknowledged that their cross-validation does not necessarily indicate good generalisation of their machine learning models.

Page 10 lines 9-10: Given that no hold-out testing was carried out, the claim of successful prediction of disease severity cannot be substantiated and should be removed or made substantially more speculative.

We agree with the points raised by the reviewer here and have added caveats to page 10. Specifically, we have added the sentence on page 10 lines 25-27: "however, it should be noted that the performance of these methods have only been demonstrated using the ISB-S datasets and may not be generalizable to other TCR repertoire datasets or for COVID-19 patients more broadly."

Reviewer 1 comments from last revision: Page 14 line 1: It would be good to know what the value of K is in the K-nearest neighbour classifier. Maybe it was set using the defaults. It would be helpful to mention whether default parameters are used in any other methods too, as they may not be appropriate for the analysis being done here.

We thank the reviewer for these suggestions. The value of K in the K-nearest neighbor classifier is $k = 3$ which we have now clarified in the methods section (page 18, which clarifies all of the hyperparameters used in our manuscript). Not all parameters are default parameters. We have attempted hyperparameter optimization of our models via a grid search of candidate parameters using the GridSearchCV function in scikit-learn's model selection library. The table of hyperparameters is shown in our last letter to reviewers.

Reviewer 1 Comments from last revision: Page 4 lines 9-10: It might be helpful to explain why different thresholds are used for different datasets.

We thank the reviewer for this comment. The reason for the different thresholds used for the AB and ISB-S datasets is due to the relative sizes of the datasets. As the AB dataset is more than an order of magnitude larger than the ISB-S dataset, we chose to use the proportion threshold of 0.0001 for ISB-S samples and 0.00001 for AB samples, so that we analyze an overall comparable number of sequences in each dataset (this explanation was included on page 6 lines 2-3).

Reviewer 2 comments

The author addressed my concerns very well. I have no more comments.

We thank reviewer 2 for their positive feedback.

Reviewer 3 comments

The underlying problems associated with this analysis remain. The datasets are highly unbalanced. The size of the repertoires will influence the diversity measurements - this has nothing to do with p-values, but arises from the distributions of TCR frequencies which are not linearly scalable. However the authors have added significant additional information and clarification so that the readers may be aware of these limitations, and take them into account in evaluating the results.

We thank the reviewer for these comments, and we acknowledge that the unbalanced nature of our datasets and their different sizes are intrinsic limitations to some of our analyses. In this revision, we have incorporated the acknowledgement of these limitations into the main body of the manuscript, which can be found in the Discussion (pages 12-13). We have also provided a table of sample counts in Figure 1A to show the differences of size sample in four datasets and add transparency. Moreover, in this revision, we have focused on incorporating the responses in the previous rebuttal letter into the body of the manuscript. We include the page and line numbers of the incorporated changes to indicate their location in the manuscript. We hope that these changes enhance the clarity of our analyses to the readers.

Reviewer 3 comments from last revision: As in all such metadata, especially where the data were produced by completely different methodologies, it is absolutely crucial to provide full clarity about the composition of the different data sets. One key element is to show how many of each cohort were mild, moderate or severe, or HD. I may have missed this somewhere in the supplementaries, but it is essential to make this data obvious in fig 1. I suspect the cohorts are very unbalanced.

We thank the reviewer for this comment. We have ensured transparency of this information by providing a table of sample counts in Figure 1 to show the differences of sample sizes in the four datasets.

Reviewer 3 comments from last revision: In fig 1, all the estimators are strongly influenced by sample size. The authors should show their conclusions are not influenced by sample size. I could not understand what panel E represented. In F, was this just a random sample of 32 AB repertoires? Why 32? Was this repeated multiple times?

We thank the reviewer for pointing this out, and we acknowledge a limitation of our study that certain conclusions, such as COVID-related TCR diversity reduction, are inevitably influenced by different sample sizes in datasets (page 12 and lines 18-21). As for Figure 1E, we have updated figure legends for clarity and we have added a subtitle in Figure 1E (CDR3 overlap between samples). Figure 1E is showing that the CDR3 sequences enriched in the COVID patients had significant overlap (among mild, moderate and severe patients), while there was almost no overlap in CDR3 enrichment between healthy donors and COVID-19 patients. This information was added to the figure legend for Figure 1E. For Figure 1F, the 32 samples are random samples from AB dataset. We clarified this in the manuscript figure legends for Figure 1F. We deemed this sample to be representative of the repertoires because repeated sampling generated very similar plots. We chose to visualize a sample of the healthy donor and COVID repertoires because it is not feasible to show properties of the entire repertoire in such a visualization due to the sheer size of the dataset.

Reviewer 3 comments from last revision: In Fig 2, the PCAs are not convincing to me. There are a few outlying mild and moderate, but otherwise they look totally overlapping. In the heatmaps, are these averages of multiple subsamples? Is there any obvious separation between condition if the samples are allowed to cluster across the different individuals? I did not really understand the import of the panels G and H. Could the authors explain this a bit more clearly?

In order to improve the clarity of the PCA results in Figure 2, we further discussed the PCA's results in the manuscript (pages 6-7). We added the following paragraph: "Because there are general rules that define many CDR3 amino acid sequences, such as the "CASS" motif that is commonly found at the beginning of many CDR3 sequences, it was expected that the majority of the data clustered together regardless of COVID infection status. The homogenous signal of shared CDR3 characteristics was likely to dominate the heterogeneous CDR3 kmers that differentiate individuals' repertoires. However, the outliers in the PCA that failed to fall into the large central cluster, which came from mild and moderate COVID samples, possibly indicated that the PCA is detecting high-variance data features that differentiate them from other CDR3 kmers, while all severe COVID samples were found within the homogenous main cluster. Machine learning is a broadly useful tool to detect and identify these fine differences in biological signal. The outliers from mild and moderate COVID patient samples may suggest that T cell repertoires may be undergoing changes that selectively enrich certain clones that harbor specific TCR motifs, in response to COVID infection, which are being captured in the PCA plot. No such changes were detected in severe COVID patients' TCR motifs in the PCA."

In the heatmaps in Figure 2B and 2D, similar to Figure 1F, we are showing the results from a random sample, of 16 patient repertoires per disease condition. It is not the average of multiple subsamples. We have clarified this in the figure legends, revising the Figure 2B and 2D legends to "Heatmaps of 3-mer abundances of a random sample of repertoires from the ISB-S CD4/CD8 dataset by disease condition (healthy donor = 16, mild = 16, moderate = 16, severe = 16)." In the visualization in Figure 2G and Figure 2H, we were interested in showing that our methods identified TCR specificity clusters that were exclusively found in COVID-19 patients, with no overlap with the TCR specificity clusters found in healthy donors. In Figure 2G we show that 677 TCR specificity clusters were found in common across the different severities of COVID-19, and in Figure 2H we show that among these 677 TCR specificity clusters, 474 were exclusive to COVID-19 and not found within healthy donors. The overall significance of Figure 2G and 2H is in showing that the motif-based prediction of antigen specificity identified TCR clusters that were exclusive to COVID-19 patients: the unique immune signatures of COVID-19.

Reviewer 3 comments from last revision: In Fig 3, what is the proportion of cells between different disease subsets? Can the authors say a bit more about the subsets? Which are naïve? Is there a proliferating subset? The authors say there is a relationship between the cluster pattern and disease severity – but it looks more like a radically different distribution between healthy and COVID-19 repertoires. This seems a striking finding, and not necessarily what other people have observed? This figure deserves a much more detailed discussion and analysis.

This paragraph on page 8 elaborates upon this discussion: “Overall, a stark contrast was observed in the clustering patterns of cells from healthy donors and COVID-19 patients in the UMAP, where cells from healthy donors were concentrated in clusters 3 and 4, while the cells from COVID patients were mostly found in cluster 6. Cluster 6 contained the proliferating subset of T cells with high degrees of clonality (**Figures 3A, S5B**), suggesting phenotypic correlates of clonal expansion. Moreover, a high density of cells in cluster 6 contained the top COVID-enriched TCR sequence motifs identified from GLIPH2 motif analysis, such as AGQGA%E, S%AAG, SL%AG, SLQGA%YE, S%SGTDT, SL%GTD, SLS%TDT, and S%AGNQP (**Figures 3C, 2F**). These clonally expanded cells containing COVID-enriched TCR sequence motifs highly expressed the gene GNLY, which encodes the cytotoxic granules of T cells, indicating that the cells in cluster 6 are primarily activated, proliferating cytotoxic T cells. We also found a correlation between clonotype expansion and COVID infection, with cells from COVID-19 patients exhibiting the highest density in effector phenotype associated cluster 6, while healthy donor cells exhibiting density in the naïve phenotype associated clusters (**Figure 3B**). We also found a higher association of lower pGen score, or private, clonotypes with cluster 6 compared to the high pGen score clonotypes (**Figure 3D**), suggesting that these clones may be specific. However, comparison of the proportion of cells for each disease condition in cluster 6 with healthy donors revealed statistically significant cell proportion increases only for the moderate condition (**Figure 3E**), despite increasing trends for all conditions. In contrast, the naïve cell subset in the UMAP plots indicated by the gene markers TCF7 and LEF1, were most abundant among healthy donors’ T cells in clusters 3 and 4, whereas few naïve T cells were observed in cluster 6 (**Figure 3B, S5D**). Altogether, these results demonstrate relationships between clonal expansion, disease status, and cell phenotype, which can be extended to subsequence motifs.”

Reviewer 3 comments from last revision: Fig 4 is lacking a lot of detail which is important for interpretation. How balanced are the repertoires? What is meant by upsampling the controls? How were the parameters decided – was this done on an independent data set (i.e. before the cross-validation)? Did the authors try fitting models on one data set, and then testing on one of the other data sets?

We thank the reviewer for this comment. In our revision, we have clarified the number of samples in each category in each of the datasets in Figure 1A of the manuscript. We have incorporated into the manuscript a description of how the parameters were decided, based on a grid search of candidate parameters using the GridSearchCV function in scikit-learn’s model selection library (method described in page 17). The Grid Search was performed on a 6mer dataset that combines CD4 and CD8 TCR repertoires of the ISB-S dataset to classify HD from all COVID sequences, so that the same hyperparameters could be used to train on CD4 and CD8 TCR data for all permutations (HD vs. Severe, HD vs. Moderate, HD vs. Mild). We have clarified the term “upsampling” in manuscript (page 17, “Following this test-train partition, to address imbalanced data, healthy donor samples were randomly resampled to be equal to the number of COVID-19 samples represented in the dataset, prior to training.”)

The parameters that were used in our final reported models are the following.

Algorithm	Final Hyperparameters (CD8 models)	Final Hyperparameters (CD4 models)
-----------	---------------------------------------	---------------------------------------

SVM	kernel='poly', C=20, degree=5, probability=True	kernel='rbf', C=100, gamma=1, probability=True
KNN	K = 3 weights='uniform' algorithm='auto' leaf_size=30 p=2 metric='minkowski'	K =3, weights='distance', algorithm='auto' leaf_size=10, p=2 metric='minkowski'
Random Forest	n_estimators=100, criterion='gini', max_depth=None, min_samples_split=2, max_features='sqrt', max_leaf_nodes=None, bootstrap=True	n_estimators = 2000, criterion='gini', max_depth = 20, min_samples_split = 5, max_features='sqrt', max_leaf_nodes=None, bootstrap=True
Bernoulli Naïve Bayes	Alpha = 1.0 Binarize = 0.0 Fit_prior = True Class_prior = None	alpha = 0, binarize = 0.0, fit_prior = True class_prior = np.array([0.5,0.5])
Gradient Boosting	n_estimators=100, learning_rate=1.0, max_depth=1	n_estimators=100, learning_rate=1.0, max_depth=20

Table 2: Final Table of Hyperparameters

Reviewer 3 comments from last revision: Overall, the attempt to provide a meta-analysis of diverse TCRrep data sets is interesting and potentially valuable. But in light of the known challenges of meta-analyses, the paper is a bit skimpy in providing detail which could be used to assess the validity of the comparisons. Also, at the end of the day, each analysis has largely been carried out on an independent data set, which rather reduces the value of carrying out a meta-analysis in the first place. Other than the overall reduction in diversity shown in fig 1, I am not convinced any of the other findings are generalizable across data sets, and so its hard to assess the significance of the findings.

We thank reviewer 3 for this feedback, and acknowledge that our manuscript has several limitations. In our revised manuscript we have elaborated upon the limitations of our study. Different aspects of these limitations are discussed throughout the paper, including page 5, which discusses the limitations of the diversity metric finding, page 6, which discusses the limitations of the PCA finding, and page 13, which discusses the limitations of conducting separate analyses on individual datasets that limits this manuscript's ability to serve as a true meta-analysis.

REVIEWERS' COMMENTS:

Reviewer #3 (Remarks to the Author):

I don't have anything much to add to my previous comments. Some significant underlying problems, raised by all the reviewers remain, and cannot really be addressed without essentially a new study. However, the authors have made a reasonable effort to highlight the limitations of the study in the text.

Response to review

REVIEWERS' COMMENTS:

Reviewer #3 (Remarks to the Author):

I don't have anything much to add to my previous comments. Some significant underlying problems, raised by all the reviewers remain, and cannot really be addressed without essentially a new study. However, the authors have made a reasonable effort to highlight the limitations of the study in the text.

We thank the reviewer for the final comment on this paper.